# Dynamic FMR1 granule phase switch instructed by m6A modification contributes to maternal RNA decay

Guoqiang Zhang[1,4], Yongru Xu[1,2,4], Xiaona Wang[2,4], Yuanxiang Zhu [1,2,4], Liangliang Wang [1], Wenxin Zhang[1], Yiru Wang[1], Yajie Gao[1,2], Xuna Wu[3], Ying Cheng [1], Qinmiao Sun [2✉] & Dahua Chen [1,2✉]

Maternal RNA degradation is critical for embryogenesis and is tightly controlled by maternal RNA-binding proteins. Fragile X mental-retardation protein (FMR1) binds target mRNAs to form ribonucleoprotein (RNP) complexes/granules that control various biological processes, including early embryogenesis. However, how FMR1 recognizes target mRNAs and how FMR1-RNP granule assembly/disassembly regulates FMR1-associated mRNAs remain elusive. Here we show that *Drosophila* FMR1 preferentially binds mRNAs containing m6A-marked "AGACU" motif with high affinity to contributes to maternal RNA degradation. The high-affinity binding largely depends on a hydrophobic network within FMR1 KH2 domain. Importantly, this binding greatly induces FMR1 granule condensation to efficiently recruit unmodified mRNAs. The degradation of maternal mRNAs then causes granule de-condensation, allowing normal embryogenesis. Our findings reveal that sequence-specific mRNAs instruct FMR1-RNP granules to undergo a dynamic phase-switch, thus contributes to maternal mRNA decay. This mechanism may represent a general principle that regulated RNP-granules control RNA processing and normal development.

[1] Institute of Biomedical Research, Yunnan University, Kunming, China. [2] State Key Laboratory of Membrane Biology, Institute of Zoology, Chinese Academy of Sciences, Beijing, China. [3] School of Life Sciences, Yunnan University, Kunming, China. [4]These authors contributed equally: Guoqiang Zhang, Yongru Xu, Xiaona Wang, Yuanxiang Zhu. ✉email: qinmiaosun@ioz.ac.cn; chendh@ynu.edu.cn

At the earliest embryonic stage, because the zygotic genomes are transcriptionally silent, early embryonic development is solely supported by maternal factors (RNAs and proteins) pre-loaded into oocytes[1]. These maternal factors are critical for early embryos but are not sufficient for the later stages of embryogenesis, during which the development is also controlled by gene products newly synthesized from the zygotic genome[2]. This change in the regulatory program is known as the maternal-to-zygotic transition[3,4]. Genetic studies have suggested that a significant proportion of maternal products is not necessary to later embryonic development and that some of these products might even be harmful to embryos at later stages[4]; the stability of maternal RNAs during early embryogenesis is therefore tightly controlled by their associated proteins. However, the molecular basis of how maternal RNA decay is initiated and how maternal RNAs are selectively cleared in a short window of time during embryonic development remain poorly understood.

Extensive studies have suggested that many RNA-binding proteins (RBPs) harbor structurally well-defined RNA-binding domains (RBDs) such as the RNA recognition motif (RRM), the K homology (KH), and DEAD-box helicase domains, as well as intrinsically disordered low complexity (LC) sequences[5,6]. The presence of RBDs and LC sequences not only allows RBPs to interact selectively with target mRNAs[6], it also regulates the dynamics of RNA granule assembly and disassembly[5,7,8]. Previous structural studies have suggested that a typical KH domain consists of approximately 70 amino acids that form a three-stranded anti-parallel β-sheet on the surface of which pack three α-helices; this structure enables RBPs to bind RNA targets with different degrees of affinity and specificity, thus exerting their functions[9,10]. Fragile X syndrome is the most common form of inherited mental retardation and is caused by the loss of Fragile X mental retardation protein1 (FMR1)[11]. FMR1 is a ribosome-associated RBP, and its selectivity for RNA targets largely depends on two central KH domains and a C-terminal (RGG) box[12,13]. In addition, the Agenet domains and the KH0 motif in FMR1 are also reported to contribute to binding RNAs[14]. Despite extensive studies of the structure and function of the FMR1 KH domains, it remains unclear how they selectively bind and regulate the protein's RNA targets[15,16]. Moreover, there is increasing evidence suggesting that the FMR1 is present within many cytoplasmic RNP granules, such as P-bodies, stress granules, and neuronal granules[8]. Recent studies have reported that FMR1 could bind the m6A modified target mRNAs, thus controlling their fate[17–20]. However, little is known about how FMR1 selectively recognizes mRNA targets and forms RNP granules to regulate RNA processing.

The m6A is the most common modification of messenger RNA (mRNA) in eukaryotic cells, and this modification affects multiple aspects of RNA metabolism, such as splicing, translation, and stability[21–23]. *Drosophila* early embryos undergo active RNA metabolisms, including maternal mRNA degradation and zygotic transcriptional activation[1,4]. In this study, we found that FMR1 preferentially binds mRNAs that contain m6A-marked "AGACU" in a residue-dependent manner. Moreover, the incorporation of m6A-modified RNAs into FMR1 granules promoted granule condensation and increased the ability of granules to efficiently sequester unmodified target maternal RNAs. Consequently, degradation of FMR1-associated mRNAs led to de-condensation of the granules, thereby ensuring proper embryogenesis. Our findings reveal the mechanism by which a subset of m6A-modified mRNAs regulates the dynamics of RNA-granules, thus contributing to the decay of target RNAs and ensuring normal development.

## Results

### The m6A controls normal embryogenesis and maternal RNA decay in *Drosophila*.

The m6A modification of mRNAs undergoes dynamic changes during *Drosophila* early embryonic development[24]. We quantified levels of m6A of mRNAs in early embryos at multiple stages by performing mass spec analysis and obtained consistent results (Fig. S1a). The dynamic changes of m6A appeared to be associated with the event of maternal RNA degradation[25] (Fig. S1b). Because the proper degradation of maternal RNAs is critical for embryogenesis, we sought to ask whether the m6A plays a role in normal embryogenesis. Given that Mettl3 and Mettl14 form the methyltransferase complex to establish m6A modification on mRNAs[26,27], we employed the CRISPR/Cas9 system[28] and generated multiple null mutant alleles of the *mettl3* and *mettl14* genes (Fig. S1c–f). To determine the maternal role of the m6A modification, we generated *mettl3* and *mettl14* single maternal mutant, and *mettl3–mettl14* double maternal mutant embryos, which displays much lower levels of m6A, compared to the wild-type control (Fig. 1a). According to the method we described previously[29], we then performed egg-hatching rate analysis and found that ~20% of embryos failed to hatch into larvae when maternal *mettl3* or *mettl14* was depleted (Fig. 1b). Moreover, nearly 40% of embryos did not reach the larval stage when maternal *mettl3* and *mettl14* were simultaneously removed (Fig. 1b), suggesting that the m6A modification is important for normal embryogenesis in *Drosophila*.

To ask whether the Mettl3–Mettl14 complex is involved in maternal RNA degradation, we generated transcriptome RNA-seq datasets of wild type and *mettl3–mettl14* double maternal mutant at multiple embryonic stages (0–0.5, 2–3, and 5–6-h stages). Based on the wild-type datasets, we clustered the mRNAs into three groups, including degraded maternal mRNAs, stable maternal mRNAs, and zygotically expressed mRNAs (Fig. 1c). By comparing the RNA-seq datasets between the wild type and *mettl3–mettl14* double maternal mutants, we found that a significant portion of maternal transcripts (686 mRNAs, $q < 0.05$) degraded in wild-type embryos was aberrantly stable in *mettl3–mettl14* mutant embryos at the 5–6-h stage (Fig. 1d and Fig. S1g, h). We next generated the m6A datasets of embryos at the 0–1-h and 5–6-h stages and identified 834 transcripts that were marked by m6A, including 380 degraded maternal transcripts, and 454 stable maternal transcripts (Fig. S1i). Interestingly, we found that the m6A signal strengths for 380 degraded maternal mRNAs were decreased markedly from the 0–1-h stage to the 5–6-h stage (Fig. S1j), whereas no apparent difference was observed in levels of m6A modification of the 454 stable maternal transcripts between the two stages (Fig. S1k). These findings suggest that the Mettl3–Mettl14 complex-mediated m6A modification contributes to normal embryogenesis and likely participates in the degradation of a portion of maternal RNAs.

### FMR1 is an m6A-RNA binding protein in *Drosophila* embryos.

YTH domain-containing proteins have been reported to function as the canonical "m6A readers" to regulate RNA fate[21,22]. For example, in zebrafish, Ythdf2 acts as an "m6A reader" to regulate the decay of a set of maternal RNAs[30]. *Drosophila* genome harbors two genes encoding YTH homologs, Ythdf and Ythdc[31]. To ask whether the YTH proteins contribute to embryogenesis, we generated a number of null mutant alleles for *ythdf* and *ythdc*, respectively (Fig. S2a–d). Genetic analysis revealed that while *ythdc* mutant flies showed phenotypes described previously (Fig. S2e), loss of maternal Ythdf or Ythdc, or double maternal Ythdf and Ythdc led to no effect on egg hatching (Fig. S2f), raising a possibility that a YTH-independent factor regulates early embryogenesis by recognizing m6A marks on mRNAs in *Drosophila*.

To search for the functional "m6A readers" specifically in the context of *Drosophila* early embryos, we synthesized a pair of biotin-

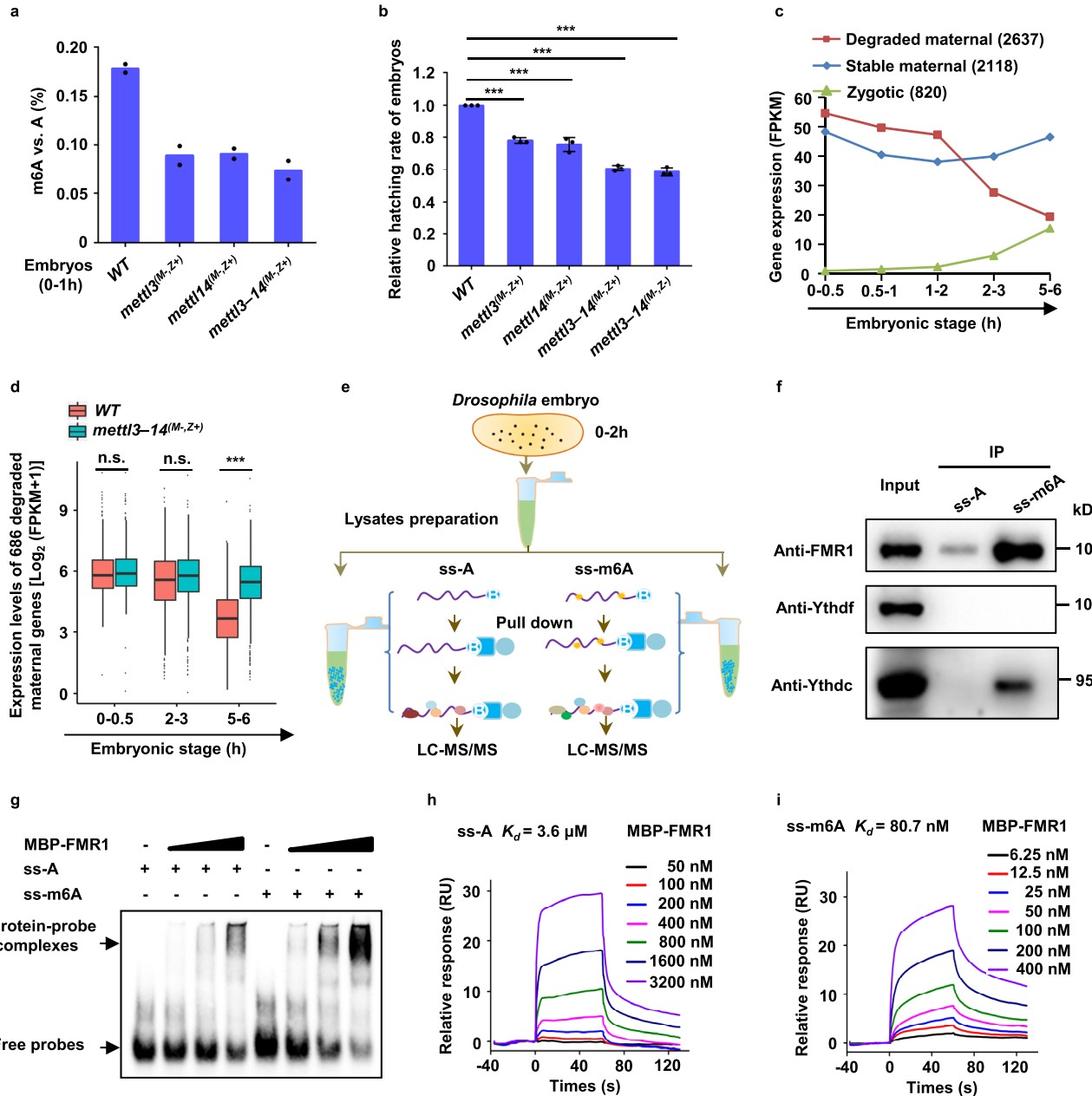

**Fig. 1 The m6A modification regulates maternal RNA decay via preferentially binding m6A-marked mRNAs. a** Abundances of m6A on mRNAs from indicated embryos were measured by mass spectrometry. The mole ratios of m6A vs. A are shown on the *y*-axis. Data expressed as means of two independent samples. **b** Relative hatching rate of embryos with indicated genotypes (*mettl3*$^{(M-,Z+)}$, $P = 2.5e-05$; *mettl14*$^{(M-,Z+)}$, $P = 0.0005$; *mettl3-14*$^{(M-,Z+)}$, $P = 4.6e-06$; *mettl3-14*$^{(M-,Z-)}$, $P = 1.6e-05$). Data expressed as means of three independent experiments. The two-sided Student's *t* test was used to analyze statistical variance. Error bars indicate mean ± SD. \*\*\**P* < 0.001. **c** Expression profile of clustered gene groups by RNA sequencing. **d** The degraded maternal mRNAs (686 overlapped mRNAs shown in Fig. S1h) in *mettl3–mettl14* double maternal mutant embryos displayed aberrant stable, when compared to wild-type at the 5–6-h stage (0–0.5 h, $P = 0.33$; 2–3 h, $P = 0.48$; 5–6 h, $P = 3e-35$). *P*-values were calculated by a one-sided Student's *t* test. The boxplot shows median (the horizontal line in the box), 1st and 3rd quartiles (lower and upper bounds of box, respectively), minimum and maximum (lower and upper whiskers, respectively). **e** Schematic for RNA pulldown and proteomics analysis. **f** Western blot assays were performed by using the anti-*Drosophila* FMR1, anti-Ythdf, and anti-Ythdc antibodies against complexes purified by the m6A modified or unmodified RNA probes. Representative figures of three independent replicates are shown. **g** EMSAs showing the complex formation of FMR1 with m6A modified or unmodified probes. Representative figures of three independent replicates are shown. **h**, **i** Surface plasmon resonance (SPR) assays of the interaction of FMR1 protein with unmodified probe (**h**) or with m6A-modified probe (**i**). Equilibrium and kinetic constants were calculated by a global fit to the 1:1 Langmuir binding model. RU resonance units. Source data are provided as a Source Data file and Supplementary Data 1 and 2.

labeled RNA probes containing multiple repeated "recognition motifs", including "GGACU", "AGACU" and "GAACU", based on the m6A recognition motifs reported in the previous studies[32,33] and the motifs identified in *Drosophila* embryos (Fig. S3a). These probes were validated by human Ythdf2 (hYthdf2, an "m6A reader" reported previously)[34] in an electrophoretic mobility shift assay (EMSA) (Fig. S3b, c). After the validation, we incubated the m6A-modified or unmodified RNA probes with cytosolic lysates from early embryos and performed immunoprecipitation experiments followed by mass spectrometry analysis (Fig. 1e). We identified 12 proteins that were either preferentially or specifically associated with m6A-modified RNA probes (Fig. S3d), and found that *Drosophila* FMRP

(FMR1) was highly abundant in the complexes immuno-precipitated by the m6A-modified probe (Fig. S3d). Further western blot assays revealed that the m6A-modified complex exhibited a much stronger FMR1 signal than the unmodified probe complex isolated from early embryos, but Ythdf was not detectable in these complexes when anti-Ythdf antibody was used (Fig. 1f and Fig. S3e). Moreover, we found that Ythdc was strongly associated with the m6A-modified probe, but not the unmodified probe (Fig. 1f). In this study, we focused on FMR1, because the loss of maternal FMR1, but not Ythdc, caused embryonic lethal phenotype. To further confirm the binding of FMR1 to m6A modified RNA, we then purified full-length FMR1 protein fused to an MBP tag from a bacterial expression system and performed EMSAs to conduct a detailed investigation of in vitro FMR1 binding to the RNA probes. EMSAs results showed that, although FMR1 could bind both the m6A-modified and the unmodified probes, signals from "shift complexes" were much stronger for the m6A-modified probe than for the unmodified probe (Fig. 1g and Fig. S3c). To quantify the binding affinities of FMR1 to the RNA probes, we performed the surface plasmon resonance (SPR) kinetic assays, found that FMR1 bound much more strongly to the m6A-RNA probe ($K_d = 80.7$ nM) than to the unmodified probe ($K_d = 3.6$ μM) (Fig.1h, i). Thus, our results suggest that FMR1 preferentially binds the m6A-modified RNAs in *Drosophila*.

**FMR1 regulates embryogenesis at least partly via binding m6A-marked RNAs**. FMR1 is required for normal embryogenesis in *Drosophila*[35–37]. To test whether FMR1 regulates early embryonic development through binding m6A-modified RNAs, we performed genetic experiments using *fmr1* null alleles (Fig. S4a), as described previously[38]. Consistent with previous findings[37], we found that more than 50% of *fmr1* maternal mutant embryos failed to reach the larval stage, and this embryonic lethal phenotype could be significantly rescued by maternal expression of FMR1 driven by the *Vas-Gal4:Vp16* driver (Fig. 2a, b). To understand the mechanism of how FMR1 regulates early embryonic development, we generated RNA-seq datasets for *fmr1* maternal mutant embryos at multiple stages (0–0.5-, 2–3-, and 5–6-h stages). By comparing these RNA-seq datasets to those from wild-type embryos, we again found that, like that in *mettl3–mettl14* double mutant embryos, a significant proportion of maternal transcripts that were degraded in wild-type embryos were aberrantly upregulated in the *fmr1* maternal mutant embryos at the 5–6-h stage (Fig. 2c and Fig. S4b, c). Intriguingly, we found that majority (~70%) of degraded maternal transcripts aberrantly upregulated in *mettl3–mettl14* double mutant embryos were also aberrantly stable in *fmr1* maternal mutant at the 5–6-h stage (Fig. S4d), suggesting that FMR1 and the m6A modification share the common mRNA targets in regulating early embryogenesis. Of note, we examined the length of these stabilized maternal mRNAs at the 5–6-h stage, and found that most of them have a length of 1000–3000 nt (Fig. S4e). To ask whether FMR1 coordinates with the m6A pathway to regulate early embryonic development, we performed a series of genetic experiments. As shown in Fig. S4f–h, eggs produced by heterozygous *fmr1* mothers, in which maternal FMR1 expression was partly reduced, fertilized by wild-type sperms displayed normal embryogenesis. The *fmr1* heterozygous embryos carrying either *mettl3* or *mettl14* maternal mutant showed much stronger embryonic lethal phenotypes, compared to maternal depletion of either *mettl3* or *mettl14* alone (Fig. 2d). These results suggested that the reduced m6A modification by depletion of either *mettl3* or *mettl14* causes embryos to be sensitive to the dose of maternal FMR1. Of note, the *fmr1* heterozygous embryos carrying *mettl3-mettl14* double maternal mutant embryos failed to enhance the embryonic lethality of *mettl3-mettl14* double maternal mutant

(Fig. 2d). Collectively, our findings support the notion that FMR1 regulates embryonic development, at least in part, through binding m6A-marked mRNAs. Of note, although YTH-domain-containing proteins are dispensable for *Drosophila* embryonic development, we do not rule out a possibility that YTH proteins could act in concert with FMR1 to regulate other biological processes. Additionally, for other potential m6A-binding factors identified in our mass spec assays (Fig. S3d), it would be interesting to examine their potential role in the future.

**FMR1 preferentially binds the m6A marked "AGACU" motif**. A long-standing yet unsolved question in the field is how FMR1 recognizes its target mRNAs with high affinity[16]. Of note, in our original screen designed to identify the m6A-modified RNA binding proteins in early embryos, we used a pair of RNA probes (51 nucleotides in length) containing repeated "GGACU", "AGACU", and GAACU" sequences with or without m6A modification. To determine which sequence contributes to the preferential binding of FMR1 with m6A-modified RNA, we synthesized additional RNA probes containing nine repeats of the "GGACU", "AGACU", "GAACU" or "AAACU" sequences, with or without m6A modification, and then performed EMSAs. As shown in Fig. 2e, the m6A-modified RNA probe containing "AGACU" sequence formed a complex with FMR1 with the highest binding affinity, whereas all unmodified probes and probes containing m6A-modified "GGACU", GAACU", or "AAACU" displayed relatively low binding affinities. To test whether the central A in "AGACU" is important for high-affinity binding of the probe to FMR1, we synthesized a mutant probe containing repeated "AGCCU" sequences and found that mutation of the central A to C markedly reduced the affinity of the RNA probe for FMR1 (Fig. S5a). We then synthesized an additional probe containing repeats of "AGA^mCU" or "A^mGACU, in which only the central A or the first A was modified. EMSAs results suggested that m6A modification in "AGACU" could support the high affinity of FMR1-probe binding (Fig. S5b). In addition, we noted that besides the "U", the "A" and the "C" are also present at the fifth position of the "RRACH" motif with high frequency that we identified in early embryos (Fig. S3a). By performing the EMSAs, we found that the probes containing m6A-modified "AGACC" and "AGAGA" motif displayed a relatively low affinity for FMR1, compared to the probe containing m6A-modified "AGACU" (Fig. S5c). Collectively, our findings suggest that the m6A-modified "AGACU" sequence is the highest affinity motif recognized and bound by FMR1.

Next, we sought to determine the mechanism by which FMR1 preferentially binds the m6A-modified "AGACU"-containing RNAs. KH domains in many RBPs interact selectively with target mRNAs in a sequence-specific manner[6,9]. We asked whether the KH domains in FMR1 contribute to its high binding affinity for mRNAs containing the specific m6A-modified motif, and found that the deletion of KH domains abolished the formation of the complex by FMR1 with the tested probes (Fig. S6a–c). Previous structural studies have suggested that the "GxxG" loop within KH domains is critical for KH-containing proteins to recognize their target RNAs[39]. To test whether the GxxG loop is required for high-affinity binding of FMR1 to the m6A-modified "AGACU"-containing RNAs, we generated an FMR1 mutant (FMR1^KH2-GDDG). EMSAs experiments revealed that the "GDDG" mutation abolished FMR1–RNA complex formation (Fig. S6d).

Selective binding of "m6A readers" to m6A-modified RNAs largely depends on a specific structure formed by certain residues in "m6A reader" proteins[40,41]. For example, in the case of hYthdf2 binding to m6A-modified RNAs, the m6A mononucleotide is

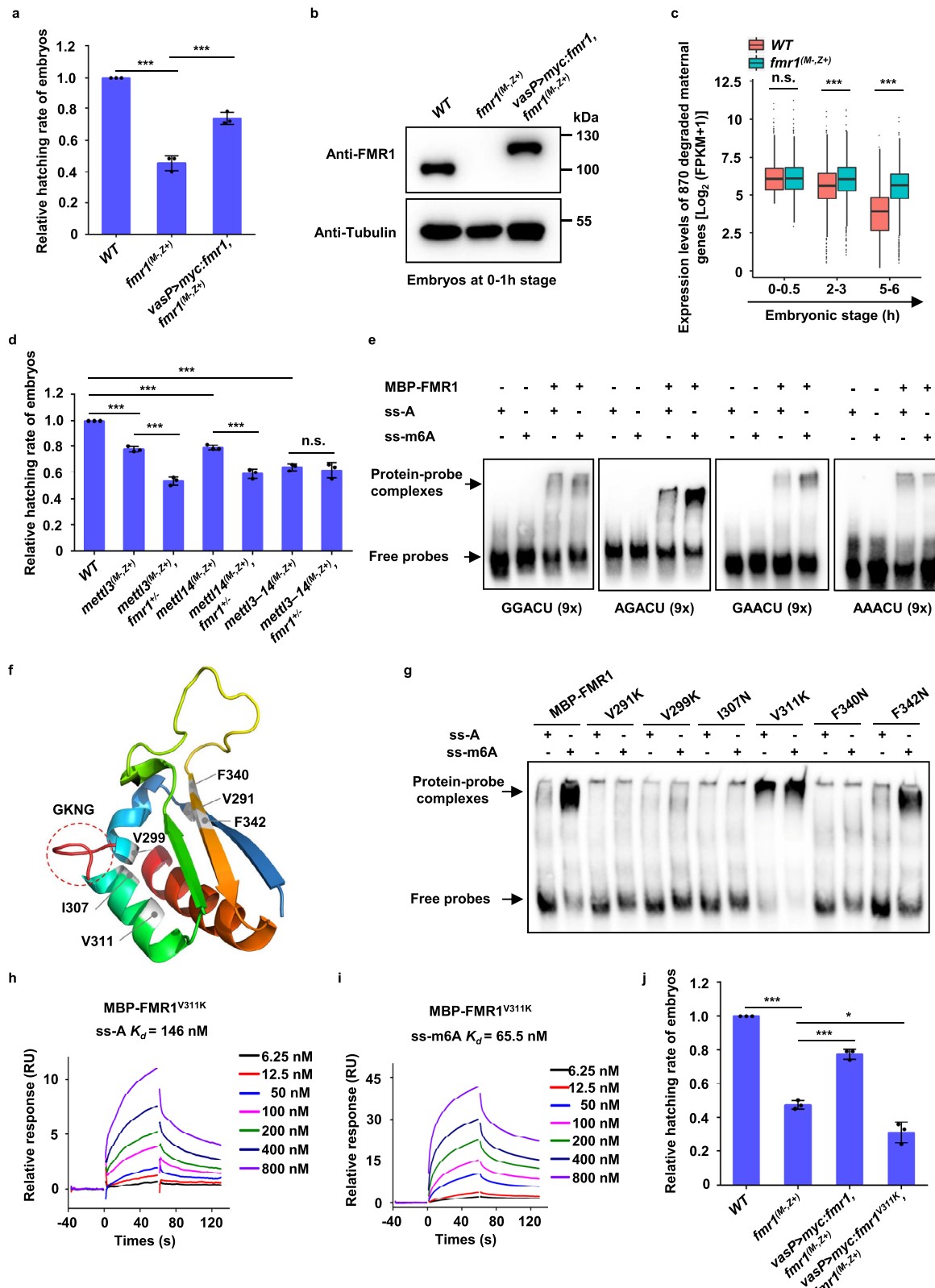

tightly locked in a hydrophobic pocket formed by residues from the α1 helix, β2 strand, and β4–β5 loop[40]. Because FMR1 can distinguish between the m6A-modified and unmodified RNAs, and given the critical role of the "GxxG" loop for FMR1 binding to RNAs, we speculated that the high affinity of FMR1 binding to RNA containing the m6A-modified "AGACU" motif might be attributable to specific residues around the "GxxG" loop. Of

interest, we noted that the KH2 domain in FMR1 contains a hydrophobic network around the "GxxG" loop, as reported in the previous structure study[42], and confirmed by our alignment and modeling analysis (Fig. 2f and Fig. S7a). Thus, we asked whether this hydrophobic network is critical for differential recognition of the m6A-modified and unmodified RNAs by FMR1. To test this possibility, we generated a series of FMR1 mutant proteins by

**Fig. 2 FMR1 preferentially binds RNAs with the m6A marked "AGACU" motif in a residue-dependent manner. a** Relative hatching rate of embryos with indicated genotypes (WT vs. $fmr1^{(M-,Z+)}$, $P = 2.6e-05$; $fmr1^{(M-,Z+)}$ vs. $vasP > myc:fmr1$, $fmr1^{(M-,Z+)}$, $P = 0.0008$). **b** Western blot assays were performed to show expression levels of endogenous FMR1 or overexpressed Myc-FMR1 in the 0–1-h embryos with indicated genotypes. The experiment was performed once. **c** The degraded maternal mRNAs (870 overlapped mRNAs shown in Fig. S4c) in $fmr1$ maternal mutant embryos displayed aberrant stable, when compared to wild-type at the 5–6-h stage (0–0.5 h, $P = 0.49$; 2–3 h, $P = 5.5e-05$; 5–6 h, $P = 3.4e-48$). $P$-values were calculated by a one-sided Student's $t$ test. The boxplot shows median (the horizontal line in the box), 1st and 3rd quartiles (lower and upper bounds of box, respectively), minimum and maximum (lower and upper whiskers, respectively). **d** Relative hatching rate of embryos with indicated genotypes (WT vs. $mettl3^{(M-,Z+)}$, $P = 4.9e-05$; WT vs. $mettl14^{(M-,Z+)}$, $P = 3e-05$; WT vs. $mettl3-14^{(M-,Z+)}$, $P = 1.9e-05$; $mettl3^{(M-,Z+)}$ vs. $mettl3^{(M-,Z+)}$, $fmr1^{+/-}$, $P = 0.0003$; $mettl14^{(M-,Z+)}$ vs. $mettl14^{(M-,Z+)}$, $fmr1^{+/-}$, $P = 0.00098$; $mettl3-14^{(M-,Z+)}$ vs. $mettl3-14^{(M-,Z+)}$, $fmr1^{+/-}$, $P = 0.56$). **e** EMSAs showing the binding capability of FMR1 to RNA probes containing either "GGACU", "AGACU", "GAACU", or "AAACU" sequences (9×) with or without m6A modification. Representative figures of three independent replicates are shown. **f** Structure modeling showing a hydrophobic network within the KH2 domain of FMR1. **g** EMSAs were performed to detect the binding affinities of FMR1 proteins carrying indicated site mutation to RNA probes with or without m6A modification. Representative figures of three independent replicates are shown. **h, i** SPR assays showing binding affinity of $FMR1^{V311K}$ protein with RNA probes with unmodified RNA probes (**h**) or with m6A-modified probes (**i**). Equilibrium and kinetic constants were calculated by a global fit to 1:1 Langmuir binding model. RU resonance units. **j** Relative hatching rate of embryos with indicated genotypes (WT vs. $fmr1^{(M-,Z+)}$, $P = 3.5e-06$; $fmr1^{(M-,Z+)}$ vs. $vasP > myc:fmr1$, $fmr1^{(M-,Z+)}$, $P = 0.0002$; $fmr1^{(M-,Z+)}$ vs. $vasP > myc:fmr1^{V311K}$, $fmr1^{(M-,Z+)}$, $P = 0.014$). In **a, d, j**, data were expressed as means of three independent experiments, and the two-sided Student's $t$ test was used to analyze statistical variance. Error bars indicate mean ± SD. *$P < 0.05$, ***$P < 0.001$. n.s. not significant. Source data are provided as a Source Data file and Supplementary Data 2.

mutating individual hydrophobic or aromatic amino acid residues around the "GxxG" loop to hydrophilic amino acids (Asn or Lys) (Fig. S7b, c). Using these mutants, we searched for the residue(s) responsible for the preferential binding of FMR1 to m6A-modified RNAs by performing an EMSA screen. As shown in Fig. 2g, while changing F342 to Asn had no apparent effect on FMR1–RNA binding, mutation of V291, V299, I307 (corresponding to I304 in human FMR1) or F340 almost completely blocked FMR1 binding to both the m6A-modified and unmodified RNA probes. Surprisingly, mutating V311 to Lys did not affect FMR1 binding to m6A-modified RNA, but significantly increased the affinity of FMR1 for the unmodified probe (Fig. 2g–i), indicating that the change of V311 residue to K affects the preferential binding of FMR1 to RNAs containing the m6A-modified "AGACU" motif. Next, we examined embryos with an expression of maternal $FMR1^{V311K}$ is driven by the $Vas$-$Gal4$:$Vp16$ driver and found that overexpression of $FMR1^{V311K}$ caused ~30% of embryonic lethal, whereas overexpression of wild type FMR1 did not affect egg hatching (Fig. S7d, e). Importantly, we found that while maternal expression of wild-type FMR1 significantly suppressed the embryonic lethal phenotype induced by loss of maternal $fmr1$, maternal expression of $FMR1^{V311K}$ further enhanced this phenotype (Fig. 2j and Fig. S7f). These findings suggest that the change of V311 to K likely affects the preferential binding of FMR1 to the m6A-modified motif in mRNAs in early embryos.

## FMR1 granules undergo a dynamical assembly in early embryos in an LC domain- and RNA binding-dependent manner.
RBPs execute their function to control the targeted RNA fate by forming RNP complexes/granules. To study the behavior of FMR1 RNP complexes/granules, we performed live-cell imaging assays to trace the FMR1–RNP complex in embryos, in which GFP-FMR1$^{FL}$ was maternally expressed. As shown in Fig. 3a, only a few FMR1 granules (less than 200 nm diameter) with weak GFP signals were present in the 0–1-h wild-type embryos, whereas many relatively larger FMR1 granules (0.5–1 μm diameter) with strong GFP signals were found in the 2–3-h embryos. Interestingly, we noted that the number of relatively large-sized granules was reduced with development, and only a few small granules were observed in the 5–6-h embryos. Similar results were obtained when an anti-FMR1 antibody was used to perform immunostaining assays (Fig. S8a–c). Thus, our findings suggest that FMR1 granules have a highly dynamic property. To obtain additional evidence to confirm this

observation, we then performed semi-denaturing detergent agarose gel electrophoresis (SDD-AGE) analysis, a method for detecting high-molecular-weight (HMW) protein aggregates[43,44]. As shown in Fig. 3b, relatively weak HMW FMR1 polymer signals were detected in wild-type embryos at the 0–1-h stage, and these signals became much stronger at the 2–4-h stage but decreased significantly at the 5–6-h stage. These findings together suggest that FMR1 granules undergo an assembly/disassembly process during early embryonic development.

We next sought to determine the requirement for FMR1 RNP-granule assembly in early embryos. LC domain has been reported to be important for many RBPs to form RNP-granules[45,46]. Given that FMR1 contains an LC domain, we, therefore, asked whether the LC domain is required for FMR1 to form RNP granules and thus regulate $Drosophila$ embryogenesis. We evaluated early embryos maternally expressing the full-length FMR1 (GFP-FMR1$^{FL}$) or a form of FMR1 lacking the LC domain (GFP-FMR1$^{\Delta LC}$). As shown in Fig. 3c, d, GFP-FMR1$^{FL}$, but not GFP-FMR1$^{\Delta LC}$, formed cytoplasmic granules in the 2–3-h embryos. Moreover, we found that maternal expression of FMR1$^{FL}$, but not FMR1$^{\Delta LC}$, significantly rescued embryonic lethal phenotypes induced by loss of maternal $fmr1$ (Fig. 3e, f), suggesting the functional importance of LC domain for FMR1 in supporting embryogenesis.

In addition to the LC domain, FMR1 also contains KH domains that enable FMR1 to selectively bind its targeted mRNAs. To test whether RNA-binding is required for FMR1 granule assembly and its dynamics, we removed the RNAs by treating the lysates at the 2-4 h embryos with RNase. By performing SDD-AGE analysis, we found that RNase treatment markedly reduced signals of HMW FMR1 polymers at the 2-4 h stage (Fig. 3g). This result indicated that the binding of RNAs to FMR1 stimulates the condensation of FMR1 granules. We then asked whether the RNA-binding activity of FMR1 contributes to dynamical assembly and disassembly of FMR1 granules during early embryonic development. Given that the "GxxG" loop is critical for FMR1 to selectively bind their target RNAs, we generated $fmr1$ maternal mutant embryos with an expression of FMR1 mutant, GFP-FMR1$^{KH2-GDDG}$, and traced the behavior of this mutant at multiple embryonic stages. As shown in Fig. 3h, i, unlike the wild-type FMR1, GFP-FMR1$^{KH2-GDDG}$ failed to form large size granules at the 2-3 h stage. Importantly, in contrast to the wild-type FMR1, maternal expression of FMR1$^{KH2-GDDG}$ was unable to rescue the embryonic lethal phenotype induced by loss of maternal FMR1 (Fig. 3j). Collectively, our results support the

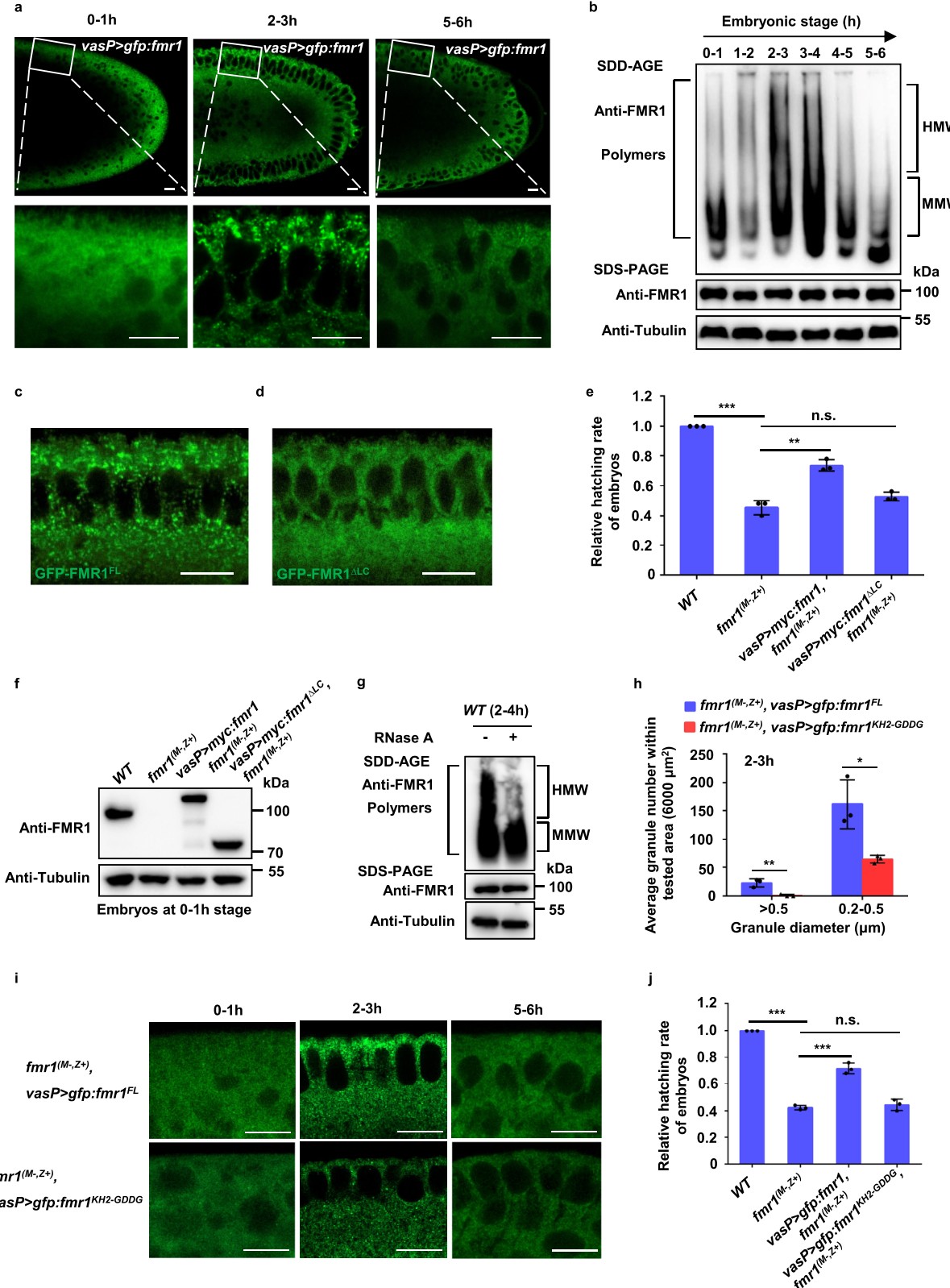

notion that specific interaction between FMR1 and RNAs dynamically regulates the assembly/disassembly switch of FMR1 granules.

**The m6A modification regulates the assembly and disassembly of FMR1 granules.** To simulate in vivo processes of FMR1 granule formation regulated by its LC domain, we purified

GFP-FMR1$^{FL}$ and GFP-FMR1$^{\Delta LC}$ (Fig. S9a), and studied the behavior of FMR1 in solution. In vitro phase separation assays revealed that GFP-FMR1$^{FL}$, but not GFP-FMR1$^{\Delta LC}$, could form micrometer-sized droplets within several minutes under low-salt conditions (110 mM NaCl) (Fig. S9b), and these small droplets then underwent fusion to form larger ones (Fig. S9c). Interestingly, we found that the droplets became smaller and finally

**Fig. 3 FMR1 granules undergo a dynamical assembly in early embryos in an LC domain- and RNA binding-dependent manner. a** Confocal imaging of live embryos with maternally expressing GFP-FMR1$^{FL}$ in wild-type embryos at the 0–1, 2–3, and 5–6-h stages. Representative figures of five independent replicates are shown. Scale bars, 10 μm. **b** Crude extracts of wide-type embryos were prepared from the indicated time points, and aliquots of the extracts were then subjected to SDD-AGE and SDS-PAGE assays. Representative figures of three independent replicates are shown. HMW high-molecular-weight, MMW middle-molecular weight. **c, d** Confocal imaging of early embryos (at the 2–3-h stage) with maternally expressing GFP-FMR1$^{FL}$ (**c**) or GFP-FMR1$^{ΔLC}$ (**d**). Representative figures of three independent replicates are shown. Scale bars, 10 μm. **e** Relative hatching rate of embryos with indicated genotypes (*WT* vs. *fmr1*$^{(M-,Z+)}$, $P = 3.3e-05$; *fmr1*$^{(M-,Z+)}$ vs. *vasP > myc:fmr1*, *fmr1*$^{(M-,Z+)}$, $P = 0.0012$; *fmr1*$^{(M-,Z+)}$ vs. *vasP > myc:fmr1*$^{ΔLC}$, *fmr1*$^{(M-,Z+)}$, $P = 0.08$). **f** Western blot assays were performed to show expression levels of endogenous FMR1, overexpressed Myc-FMR1 or Myc-FMR1$^{ΔLC}$ in the 0–1-h embryos with indicated genotypes. The experiment was performed once. **g** Crude extracts of wide-type embryos by treating with RNase A were prepared, and aliquots of the extracts were then subjected to SDD-AGE and SDS-PAGE assays. Representative figures of two independent replicates are shown. HMW high-molecular-weight, MMW middle-molecular weight. **h** Quantifying the average number of FMR1 granules with different sizes within tested area (6000 μm$^2$) in (**i**) (>0.5, $P = 0.0062$; 0.2–0.5, $P = 0.018$). **i** Confocal imaging of live embryos with maternally expressing GFP-FMR1$^{FL}$ or GFP-FMR1$^{KH2-GDDG}$ in *fmr1* maternal mutant embryos at the 0–1, 2–3, and 5–6-h stages. Scale bars, 10 μm. **j** Relative hatching rate of embryos with indicated genotypes (*WT* vs. *fmr1*$^{(M-,Z+)}$, $P = 3.3e-07$; *fmr1*$^{(M-,Z+)}$ vs. *vasP > gfp:fmr1*, *fmr1*$^{(M-,Z+)}$, $P = 0.0003$; *fmr1*$^{(M-,Z+)}$ vs. *vasP > gfp:fmr1*$^{KH2-GDDG}$, *fmr1*$^{(M-,Z+)}$, $P = 0.47$). In **e**, **h**, **j**, data were expressed as means of three independent experiments, and the two-sided Student's *t* test was used to analyze statistical variance. Error bars indicate mean ± SD. *$P < 0.05$, **$P < 0.01$, ***$P < 0.001$. n.s. not significant. Source data are provided as a Source Data file.

disappeared, when salt concentrations were increased (Fig. S9d), suggesting that FMR1 droplet formation is a reversible process regulated by phase separation.

Given that RNA binding stimulates FMR1 granule assembly and controls the dynamic of FMR1 granules in embryos, and that FMR1 has a high affinity for the m6A-modified "AGACU" motif-containing RNAs, we speculated that FMR1 binding to the m6A-modified "AGACU" probe might influence dynamics of FMR1 droplets. To test this possibility, we incubated the GFP-FMR1$^{FL}$ with the m6A-modified "AGACU" probe or the unmodified probe. As shown in Fig. 4a–c and Fig. S9e, the m6A-modified probe not only co-phase separated with GFP-FMR1$^{FL}$ (but not GFP-FMR1$^{ΔLC}$), it also significantly increased the size of the FMR1$^{FL}$-RNA droplets. By contrast, the unmodified RNA probes only displayed a weak co-phase separation with FMR1$^{FL}$, and this weak incorporation did not apparently change the size of FMR1–RNA granules (Fig. 4a–c and Fig. S9e). Of note, neither the m6A-modified nor the unmodified probe alone showed any sign of phase separation (Fig. S9e). We next performed fluorescence recovery after photo-bleaching analysis and found that the FMR1–m6A-modified RNA droplets showed minimal recovery after photo-bleaching (Fig. 4d, e), compared with FMR1-unmodified RNA droplets or droplets containing only FMR1 without mixed with RNAs, suggesting that binding to m6A-modified RNA significantly changes the FMR1 condensate fluidity.

Of note, based on our datasets, loss of maternal FMR1 not only affected m6A-marked mRNA degradation, but it also caused the decay of a portion of non-m6A-marked mRNAs in early embryos. Similar results were observed in *mettl3–mettl14* double maternal mutant embryos. We reasoned that the incorporation of m6A-marked RNAs might affect the ability of FMR1 granules to further sequester unmodified RNAs. We thus performed additional in vitro phase separation assays. As shown in Fig. 4f, g, the presence of m6A-marked RNAs in solution induced much stronger co-phase separation of FMR1 and unmodified RNAs, compared with the absence of m6A-marked RNAs. Of note, when we used the negative control RNA probes that did not bind the FMR1 protein (Fig. S9f), we found that these probes failed to be recruited into FMR1 droplets, even in the presence of m6A-marked RNAs containing "AGACU" motif (Fig. 4h). These findings provide an important line of evidence indicating that the presence of m6A-modified RNA increases the size of FMR1 granules and enhances the partitioning of non-methylated RNA into the granules. To gain in vivo evidence for supporting our conclusion, we performed live-cell imaging assays to trace the behavior of GFP-FMR1$^{FL}$ in *mettl3–mettl14* double maternal

mutant embryos and found that number of the relatively large-sized granules was significantly reduced (Fig. 4i and Fig. S9g, h). In support of this, the SDD-AGE assay showed a dynamic change of high molecular weight of FMR1 in wild-type and *mettl3–mettl14* maternal embryos from different developmental stages (Fig. S9i). Collectively, these findings emphasize the importance of m6A modification in enhancing FMR1 granule size and partitioning of RNA molecules into the granules.

**FMR1 regulates decay of its target maternal mRNAs in an m6A-dependent manner.** FMR1 has been previously proposed to repress target gene expression through translational repression[47,48]. In this study, we found that loss of maternal FMR1 led to the aberrant stability of a portion of the degraded maternal mRNAs. Thus, there are two possibilities that can explain our results: (1) FMR1 directly regulates control target gene expression through RNA decay; (2) aberrant stable of maternal mRNAs induced by loss of maternal FMR1 could be a consequence (a byproduct) of defective translational repression. To distinguish these two possibilities, we performed data-independent acquisition (DIA) quantitative proteomics on the wild-type and *fmr1* maternal mutant embryos at multiple developmental stages, including 0–1-, 2–3-, and 5–6-h stages (each sample with three biological replicates) (Fig. S10a). We identified 4987 protein hits (high confident hits) shared with each sample. It is worth noting that if the "byproduct" mode were correct, aberrant stable of degraded maternal mRNAs should be highly associated with up-regulation of corresponding protein products. To address this important issue, we analyzed the proteomic datasets together with RNA-seq datasets and identified the 4814 hits (Fig. S10b). Because FMR1 is a well-known translation repressor and potentially regulates RNA decay, we then focused on analyzing the upregulated genes and no significant changed genes at either mRNA or protein levels. As shown in Fig. 5a and Fig. S10c, we classed these genes into three major groups: genes in group 1 showed no significant change at both protein and mRNA levels; genes in group 2 displayed an increase at protein levels, but no significant change in levels of mRNAs; and genes in the group 3 showed an increase at mRNA levels, but no significant change in levels of proteins. Interestingly, we found that the majority of targets with aberrant upregulation at mRNA levels showed a decrease or no change in levels of proteins (group 3), suggesting that FMR1 has a role in regulating maternal mRNA decay. Additionally, we also found that genes in group 3 were significantly correlated with m6A modification, when compared to group 2 (Fig. 5b). Collectively, our findings suggest that in addition to translation repression,

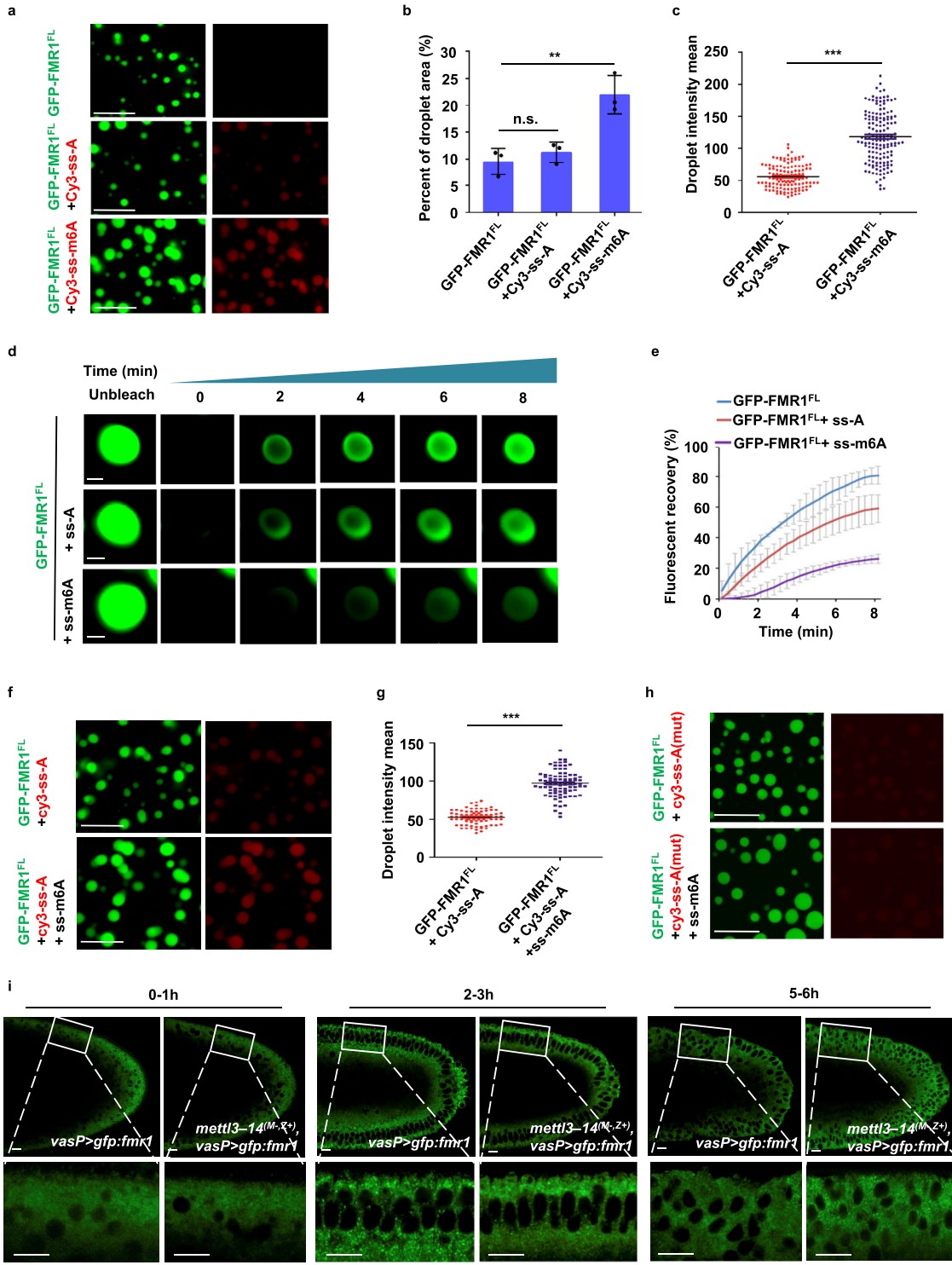

FMR1 indeed plays a role in maternal mRNA decay through m6A modification.

To determine the mechanism of how FMR1 regulates maternal RNA decay, we collected the 2–3-h embryos and performed immunoprecipitation using the anti-*Drosophila* FMR1 antibody, followed by mass spectrometric analysis. As shown in Fig. S10d, in addition to ribosomal proteins and the Caprin protein, which are known FMR1-associated partners, poly(A)-binding protein (pAbp) was also highly enriched in the FMR1 complex in early embryos. pAbp has been shown to be involved in mRNA deadenylation, thus contributing to mRNA degradation[49,50].

We reasoned that interaction between FMR1 and pAbp could provide a mechanistic explanation for the action of FMR1 to regulate mRNA decay. In order to address the issue, we performed immunostaining and co-immunoprecipitation (co-IP) experiments. Immunostaining assays showed that pAbp was significantly overlapped with a portion of condensed FMR1 granules with a high Pearson's correlation coefficient ($r = 0.81$) (Fig. 5c, d). The co-IP analysis suggested that FMR1 and pAbp could form a complex in wild-type early embryos (the 2–4-h stage) (Fig. S10e), suggesting that pAbp is associated with condensed FMR1 granules. Interestingly, the further co-IP

**Fig. 4 The m6A modification regulates assembly/disassembly switch of FMR1 granules. a** Droplet formation assays for GFP-FMR1$^{FL}$(10 μM) mixed with Cy3-labeled m6A-modified or unmodified probes, or alone. Scale bars, 10 μm. **b** The quantitative area of green droplet signal in (**a**) (GFP-FMR1$^{FL}$ + Cy3-ss-A, $P = 0.41$; GFP-FMR1$^{FL}$ + Cy3-ss-m6A, $P = 0.008$). **c** Quantitative intensity of RNA probes (red) droplet signal in (**a**) ($P = 1.7e{-}40$). **d** Fluorescence recovery after photo-bleaching (FRAP) assays to study the in vitro behavior of FMR1 droplets. Representative figures of four independent replicates are shown. Scale bars, 1 μm. **e** Changes in the fluorescence intensity of droplets after photobleaching were plotted over time. The curve represents the mean of the fluorescence intensity in the photobleached region of interest in distinct droplets ($n = 4$). Error bars indicate mean ± SD. **f** Droplet formation assays for the ability of GFP-FMR1$^{FL}$(10 μM) to sequester unmodified RNAs (Cy3-labeled) with m6A-modified RNA probes or alone. Scale bars, 10 μm. **g** Quantitative intensity of RNA probes (red) droplet signal in (**f**) ($P = 4.4e{-}51$). **h** Droplet formation assays for testing the ability of GFP-FMR1$^{FL}$ (10 μM) to sequester unmodified mutant RNAs (Cy3-labeled) with m6A-modified RNA probes or alone. Representative figures of three independent replicates are shown. Scale bars, 10 μm. **i** Confocal imaging of live embryos with maternally expressing GFP-FMR1$^{FL}$ in wild-type or *mettl3-mettl14* double maternal mutant embryos at the 0–1, 2–3, and 5–6-h stages. Representative figures of three independent replicates are shown. Scale bars, 10 μm. In **b**, **c**, **g**, data were expressed as means of three independent experiments, and the two-sided Student's *t* test was used to analyze statistical variance. Error bars indicate mean ± SD (**b**) or SEM (**c**, **g**). **$P < 0.01$, ***$P < 0.001$. n.s. not significant. Source data are provided as a Source Data file.

analysis showed that loss of maternal Mettl3–Mettl14 markedly reduced the association of FMR1 and pAbp (Fig. 5e). As mentioned above, pAbp affects mRNA stability through mRNA polyadenylation/de-adenylation[49,50], we reasoned that aberrant upregulation of maternal mRNAs induced by loss of maternal FMR1 could be attributable to inefficient deadenylation. To test this possibility, we measured the poly(A) tail length of FMR1 targets, such as *RASSF8*, *hdc*, and *HIP* in wild-type and *fmr1* maternal mutant embryos at multiple stages. We chose these targets for the further assays based on two observations. First, these targets contain the m6A-modified "AGACU" motif (Fig. S11a–c) and undergo degradation during early embryonic development, as tested by q-RT-PCR (Fig. S11d–f). Second, by using anti-FMR1 antibody, we performed RNA-IP followed by q-RT-PCR assays and found that these targets were indeed associated with FMR1 (Fig. S11g). By carrying out poly(A) tail length (PAT) assays, we found that *RASSF8*, *hdc*, and *HIP* transcripts underwent normal deadenylation in wild-type embryos, but were less deadenylated in *fmr1* maternal mutants (Fig. 5f). These findings further suggest that loss of maternal *fmr1* leading to aberrant up-regulation of these m6A-modified transcripts was due to inefficient deadenylation. Moreover, we obtained similar results, when these targets were analyzed in the *mettl3–mettl14* double maternal mutant embryos (Fig. 5f). Interestingly, in agreement with our argument that binding of m6A-modified RNAs efficiently sequesters unmodified RNAs into FMR1 granules, we found that FMR1 regulated genes (*Crag*, *Gdh*, and *Dah*) without m6A modification, which underwent degradation (Fig. S11h–j) were also less deadenylated in embryos when maternal FMR1 was depleted (Fig. S11k).

## Discussion

During early embryogenesis, a significant proportion of the maternal transcriptome is degraded and gradually replaced by zygotic transcripts, thus allowing normal embryonic development[1]. However, little is known about how maternal RNA decay is triggered and how maternal RNAs (but not zygotic RNAs) are selectively cleared by RNP complexes. In this study, we report that a subset of m6A-modified mRNAs instructs FMR1–RNP granules to undergo a dynamic phase-switch, thus contributing to maternal RNA degradation during early embryogenesis. We show that while the LC domain initiates FMR1 granule self-assembly, KH domains facilitate the high-affinity binding of FMR1 to the mRNAs containing m6A-modified "AGACU" motif. This binding leads to significant condensation of FMR1 granules and increases the granules' ability to sequester unmodified RNAs. Accompanying with degradation of the maternal mRNAs including m6A-modified RNAs, the FMR1 granules then undergo a de-condensation process, which allows normal embryogenesis to proceed (Fig. 5g). Our findings demonstrate

the mechanism by which a subset of sequence-specific RNAs regulates a functional dynamical phase-switch of certain RNP–granules, thus contributing to RNA processing and normal development.

**FMR1 regulates a subset of maternal mRNA decay, at least in part, via m6A.** It has been reported previously that FMR1 and Caprin form an RNP-complex to control *Drosophila* early embryogenesis by directly mediating the translational repression of maternal *cycB*, suggesting a role of the FMR1/Caprin complex in early embryos through translation control[36]. By performing the hatching-rate analysis, we found that loss of maternal Caprin led to relatively weak phenotypes (~20% of embryonic lethality) (Fig. S12a–c). Given that the loss of maternal FMR1 caused a relatively strong phenotype (~55% of embryonic lethality), our results suggest that FMR1 may have other functions to control embryogenesis, in addition to its role via Caprin.

Given that maternal RNA decay is a hallmark of early embryogenesis, in this study, we investigate whether FMR1, as a typical RBP, contributes to maternal RNA decay in *Drosophila* early embryos. Several lines of evidence support that FMR1 is involved in maternal RNA decay. First, RNA-seq analyses showed that the loss of maternal FMR1 led to aberrant upregulation of a portion of the degraded maternal RNAs in early embryos. Second, loss of maternal Mettl3–Mettl14 also caused aberrant stability of a portion of the degraded maternal RNAs. Interestingly, we found that aberrantly upregulated maternal RNAs from *fmr1* maternal mutant embryos were highly overlapped with those from *mettl3–mettl14* mutant maternal embryos, suggesting that FMR1 and the m6A pathway share common targets in early embryos. Third, by performing DIA mass spec experiments, we found that loss of maternal FMR1 caused a defect of RNA decay in addition to translational repression. In order to link FMR1 to the deadenylation pathway, we performed immunostaining and co-immunoprecipitation experiments. Our immunostaining assays showed that signals of the pAbp protein were significantly detected in a portion of condensed FMR1 granules. The co-IP analysis suggested that FMR1 and pAbp could form a complex in wild-type early embryos (the 2–3-h stage), suggesting that pAbp is associated with condensed FMR1 granules. These findings together support the notion that FMR1 regulates maternal RNA decay, at least in part, via m6A modification.

**How does FMR1 preferentially bind the "AGA$^m$CU" motif in maternal mRNA?** It has been reported that several missense mutations within the KH domains in FMR1 are associated with FXS, suggesting that FMR1 KH domains play roles in the pathophysiology of FXS[12,51,52]. Indeed, in this study we found that KH domains are essential for FMR1–RNA binding and critical for preferential binding of FMR1 to m6A-modified RNAs in

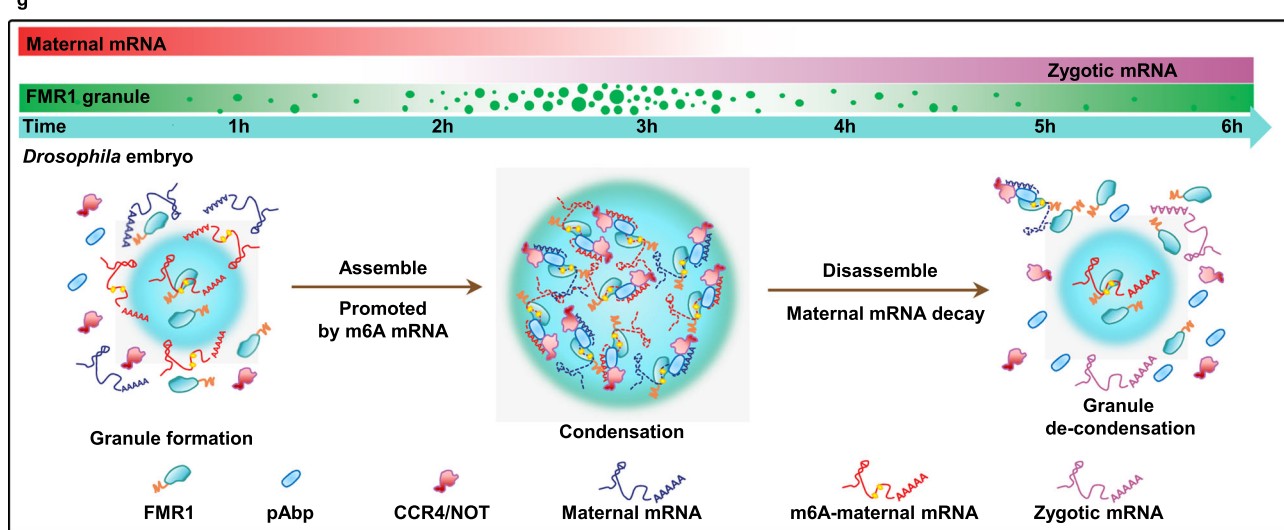

**a** 5-6 h

Group 2 (mRNA no significant change, protein up)

Group 1 (mRNA and protein no significant change)

Group 3 (mRNA up, protein no significant change or down)

**b** 5-6 h

- Group 2 without m6A
- Group 2 with m6A (P = 0.93)
- Group 3 without m6A
- Group 3 with m6A (P = 0.0066)

**c** WT — FMR1, pAbp, Merge

**d** r=0.81 — FMR1 pAbp

**e** Lysate — Anti-FMR1 (100 kDa), Anti-pAbp (70 kDa); IP: FMR1 — Anti-FMR1 (100), Anti-pAbp (70); Embryos at 2-4h stage

**f** Embryos (h): WT 0-1 5-6, fmr1(M-,Z+) 0-1 5-6, mettl3-14(M-,Z+) 0-1 5-6, Size (bp) — RASSF8, hdc, HIP, β tub56D

**g** Maternal mRNA — Zygotic mRNA; FMR1 granule; Time 1h 2h 3h 4h 5h 6h; Drosophila embryo; Granule formation → Assemble Promoted by m6A mRNA → Condensation → Disassemble Maternal mRNA decay → Granule de-condensation; FMR1, pAbp, CCR4/NOT, Maternal mRNA, m6A-maternal mRNA, Zygotic mRNA

a sequence-specific manner. Consistent with previous findings[39], we found that the "GxxG" loop in KH domains is critical for FMR1 to interact with RNA targets. Structural studies of hYthdf2 have suggested that the m6A mononucleotide is tightly locked in a hydrophobic pocket[40]. In this study, we tested whether the preferential recognition of m6A-modified RNA by FMR1 can be attributed to specific residues around the "GxxG" loop. By performing alignment and modeling analysis, we found that *Drosophila* FMR1 contains a hydrophobic network around the "GxxG" loop. Through an EMSA screen, we found that the change of the

V311 residue to K affects the preferential binding of FMR1 to m6A modified RNA targets. Thus, our findings suggest that the high-affinity FMR1 binding to the specific m6A-modified motif might require the presence of a single specific residue in FMR1.

**Phase-switch of FMR1 granules contributes to maternal RNA decay.** Of note, FMR1 not only binds to m6A-modified RNA with high affinity, but it also associates with unmodified RNA with low affinities in vitro, thus the question becomes what the biological

**Fig. 5 FMR1 regulates the decay of its target maternal mRNAs in an m6A-dependent manner. a** Scatterplot integrating protein and mRNA levels were generated by using wild-type and *fmr1* maternal mutant embryos at the 5–6-h stage. By comparing fold change of *fmr1* maternal mutant versus wild-type embryos, three major groups were classified: Group 1 (gray), including the genes without any change in both protein and mRNA level; Group 2 (blue), including the genes with upregulated protein level, but limited mRNA level change; Group 3 (red), including the genes with upregulated mRNA level, but limited or decreased protein level change. **b** Group 2 and Group 3 were further analyzed to overlap with m6A marked genes. We identified that 537 out of 4814 genes (used for proteomic and transcriptomic analyses) were marked with m6A. In total, m6A modification (highlighted with black circle) was found in as few as 5 genes in Group 2 (two-sided chi-square test, $P = 0.93$), but in a significant portion of genes (116) in Group 3 (two-sided chi-square test, $P = 0.0066$). **c** Wild-type embryos (at the 2–3-h stage) were stained with anti-FMR1 (red) and anti-pAbp (green) antibodies. Representative figures of three independent experiments are shown. Scale bars, 10 μm. **d** Pearson's correlation coefficients of line profiles of FMR1 and the indicated proteins ($n = 10$). Error bars indicate mean ± SEM. **e** Coimmunoprecipitation of FMR1 with pAbp in wide-type and *mettl3–mettl14* double maternal mutants at 2–4-h embryonic stage. The lysates were immunoprecipitated with anti-FMR1 antibody, and western blot assays were performed to detect pAbp protein in each immunoprecipitation. Representative figures of two independent replicates are shown. **f** PAT assay results showing changes in poly(A)-tail length for the indicated transcripts in the 0–1 and 5–6-h embryos with indicated genotypes. Representative figures of three independent replicates are shown. **g** Model for dynamic condensation/de-condensation of FMR1 granules to regulate maternal mRNA decay in fly early embryos. Source data are provided as a Source Data file and Supplementary Data 4.

significance of the differential binding of FMR1 to target RNAs is. In addition to other RNA-binding domains, FMR1 also contains a large LC domain at its C-terminus[5,53]. We showed that the LC domain and KH domains coordinate to regulate FMR1 granule condensation and de-condensation in *Drosophila* early embryos through high-affinity binding of FMR1 with m6A-modified RNAs and the dynamically regulated m6A modification, respectively. Thus, the dynamic phase-switch likely confers the function of FMR1 granules in selectively regulating the degradation of maternal mRNA, but not newly synthesized zygotic transcripts in the later stage embryos, thus ensuring proper embryogenesis. These findings support a model that a subset of sequence-specific RNAs can instruct a symmetric phase-switch of FMR1 granules, thus regulating maternal RNA decay in early embryos.

## Methods

**Drosophila strains**. Fly stocks used in this study were maintained under standard culture conditions. The *Drosophila* $w^{1118}$ strain was used for collecting the wild-type embryos and the host for P element-mediated transformations in this study. The *fmr1* null flies, $fmr1^{\Delta 50}$, and $fmr1^{\Delta 113}$ were described previously[38]. The following fly strains were generated in this study: (1) The mutant strains: $mettl3^1$, $mettl3^2$, $mettl14^1$, $mettl14^2$, $ythdf^1$, $ythdf^2$, $ythdc^1$, $ythdc^2$, and $caprin^1$ mutant alleles, were generated according to the method described previously[54]. Briefly, the sequences targeted by gRNAs were chosen according to the CRISPER/Cas9 target software (http://zifit.partners.org/ZiFiT). The gRNA was in vitro transcribed and injected into embryos expressing *Cas9* driven by the *vasa* promoter. The mutant alleles were screened and finally identified by sequencing; (2) the transgenic strains include P{uasp-myc-fmr1}, P{uasp-myc-fmr1$^{V311K}$}, P{uasp-myc-fmr1$^{\Delta LC}$}, P{uasp-gfp-fmr1}, P{uasp-gfp-fmr1$^{\Delta LC}$}, and P{uasp-gfp-fmr1 $^{KH2-GDDG}$}, in which the full-length, mutant or truncated *fmr1* tagged by *myc* or *gfp* were placed under the control of the UAS promoter; (3) P{vasa-gal4:vp16} is a knock-in strain, in which the *gal4:vp16* sequence was placed at the downstream of the *vasa* promoter. This strain was used as a maternal driver. The detailed information of primers is described in Supplementary Table 1.

**Embryo preparation**. *Drosophila* embryo samples were collected according to the method described recently[29]. Briefly, the well-fed flies were used to lay eggs in bottles, each of which was covered by a petri dish with agar gel. Embryos were carefully collected and examined under light microscopes. The older embryos were removed using the light microscope and the embryos were cleaned with washing buffer (1× PBS) to avoid contamination. Then the samples were immediately frozen in liquid nitrogen and stored at −80 °C.

**RNA sequencing (RNA-seq)**. Total RNAs were isolated from embryos at the indicated stages using TRIzol reagent (ThermoFisher, 15596026). The mRNAs were purified using a NEBNext Poly(A) mRNA Magnetic Isolation Module Kit (NEB, E7490) according to the user manual. Illumina Truseq libraries were constructed according to the manufacturer's instructions and sequenced using NovaSeq platform.

**MeRIP-m6A-Seq**. Firstly, 5 μg of fragmented mRNAs were incubated with 5 μg of the anti-m6A polyclonal antibody (1:100; Millipore, ABE572) in IP buffer (150 mM NaCl, 10 mM Tris-HCl, 0.1% NP-40, pH 7.4) in two independent biological replicates for 2 h at 4 °C. The mixture was incubated with Pierce protein A/G

agarose (ThermoFisher, 20421) at 4 °C for an additional 2 h. Beads with captured RNA fragments were then immediately washed 3 times with ice-cold IP buffer. Then, the bound mRNAs were eluted from the beads with N6-methyladenosine (Berry & Associates, 1867-73-8) in IP buffer, and were extracted with Trizol reagent. Finally, both of the input samples and the m6A-IPed samples were prepared for RNA-seq library construction using the NEBNext® Ultra II Directional RNA Library Prep Kit (NEB, E7760). Library sequencing was subsequently performed on an Illumina Hiseq 4000 sequencer with 150 bp paired-end reads.

**m6A quantification by UHPLC–MRM–MS/MS**. The polyadenylated RNA from indicated embryos was isolated using the NEBNext Poly(A) mRNA Magnetic Isolation Module Kit (NEB, E7490). In brief, 100 ng isolated mRNAs were digested by nuclease P1 (NEB, M0660S) in 40 μl buffer containing 10 mM $NH_4OAc$ (pH 5.3) at 42 °C for 6 h, followed by the addition of $NH_4HCO_3$ and alkaline phosphatase (NEB, M0525S). After an additional incubation at 37 °C for 6 h, the solution was diluted with dd $H_2O$ to 200 μl final volume. Then, 10 μl of the solution was subjected to the LC-MS/MS analysis. The nucleosides were separated by reverse-phase ultra-performance liquid chromatography on a C18 column and were detected using Waters TQ-S QQQ triple quadrupole LC–MS in positive electrospray ionization mode. The nucleosides were quantified using the nucleoside to base ion mass transitions of 282-150 (m6A), and 268-136 (A). Quantification was performed by comparison with the standard curve obtained from pure nucleoside standards running at the same batch of samples. The ratio of m6A to A was calculated based on the calculated concentrations.

**Analysis of hatching rate**. Embryos were collected for 24 h at 25 °C, then removed from the adults and allowed to develop for another 36 h. Then, unhatched embryos and larvae were calculated for the hatching rate.

**Antibody preparation and western blot analysis**. The antibodies were generated by immunizing mice or rabbits with recombinant proteins that were produced in *Escherichia coli*. The recombinant proteins were used as follows: GST-Mettl3 (amino acids 1–200 aa), GST-Mettl14 (amino acids 52–202 aa), GST-Ythdf (amino acids 501–700 aa), MBP-Ythdc (amino acids 351–721 aa), MBP-pAbp, and MBP-Caprin (amino acids 351–600 aa). Western blots were performed by using standard protocols. The antibodies were used as follows: mouse anti-β-Tubulin (1:2000; Cwbio, CW0098M); rabbit anti-Myc (1:2000; MBL, 562); mouse anti-*Drosophila* FMR1 (1:2000; Abcam, ab10299); mouse anti-Mettl3 (1:1000); mouse anti-Mettl14 (1:1000); mouse anti-Ythdf (1:1000); mouse anti-Ythdc (1:1000); rabbit anti-pAbp (1:2000); rabbit anti-Caprin (1:2000). The quantitation of band intensity was measured using ImageJ (1.46r) software.

**Recombinant protein expression and purification**. All recombinant proteins for EMSA and phase separation assay were expressed and purified from *Escherichia coli* BL21(DE3) using pET28a vector. In brief, The MBP or GFP tag was added on the N terminus of FMR1 to the constructs. BL21 cells were grown in LB medium supplemented with 50 μg/ml kanamycin at 37 °C to an optical density (OD600 = 0.6). Then cells were induced with 0.5 mM IPTG for 18 h at 16 °C. Cells were harvested at 1500*g* for 5 min at 4 °C and resuspended in lysis buffer (20 mM Tris-Cl, pH8.0, 500 mM NaCl, 5 mM imidazole, 1% TritonX-100). The cells were lysed by sonication and then centrifuged at 15,800 g for 20 min. The supernatants were collected and purified using Ni Sepharose™ High-Performance beads (GE Healthcare, 17-5268-01). The beads were washed twice with wash buffer 1 (20 mM Tris-Cl, pH8.0, 500 mM NaCl, 20 mM imidazole) and once with wash buffer 2 (20 mM Tris-Cl, pH8.0, 500 mM NaCl, 50 mM imidazole). The recombinant proteins were eluted using elution buffer (20 mM Tris-Cl, pH8.0, 500 mM NaCl,

500 mM imidazole, 10% glycerol). All protein purification steps were performed at 4 °C.

**Synthesis of RNA probes**. Biotin/Cy3-labeled A/m6A ssRNA probes were synthesized using Riboprobe® in vitro Transcription Systems (Promega, P1440). Briefly, a double-stranded DNA template at a final concentration of 100 nM was incubated with ATP or m6ATP (Trilink, N-1013), GTP, CTP, Biotin-16-UTP (Roche, 11388908910) or Aminoallyl-UTP-Cy3 (Jena, NU-821-CY3), and other reaction components in a 20 µl reaction volume following the kit's instructions. The RNA products were purified by ethanol precipitation. 5′-biotinylated "AmGACU" and "AGAmCU" RNA probes, in which only the first A or the central A was modified by m6A, were synthesized by RiboBio Biotechnology Company. The primers used for the generation of DNA templates are shown in Supplementary Table 2.

**Electrophoretic mobility shift assays (EMSAs)**. In vitro synthesized biotin-A/m6A ssRNA probes and purified proteins were used to perform this assay. Briefly, the indicated recombinant proteins were expressed in *E. coli* and purified by His-tag affinity purification. The biotin-A/m6A ssRNA probes were incubated with purified protein in 10 µl binding buffer (10 mM HEPES, pH 7.4, 50 mM KCl, 1 mM DDT, 1 mM EDTA, 0.05% TritonX-100, 5% glycerol, 10 ng/µl salmon DNA, and 2 U/ml ribonuclease inhibitor) at 25 °C for 30 min. Then, 1 µl glutaraldehyde (0.2% final concentration) was added to the mixture and incubated at 25 °C for 15 min. Totally, 11 µl RNA-protein mixture was subjected to 6% native polyacrylamide gel and run for 40 min at 90 V in 0.5× TBE buffer. The gel was then transferred to a positively charged nylon transfer membrane (GE Healthcare, RPN303B) and the membrane was cross-linked by UV irradiation (120 mJ/cm$^2$) using a commercial UV-light crosslinking instrument. The membrane was blocked with blocking buffer (Beyotime, GS009B) and subsequently incubated with Streptavidin HRP (1:5000; Abcam, ab7403) diluted by blocking buffer for 15 min. The membrane was washed with washing buffer (Beyotime, GS009W), and incubated with equilibration buffer (Beyotime, GS009A) for 5 min. The immunoreactive bands were visualized using an ECL system.

**Biotinylated RNA pull-down assay followed by the LC–MS/MS analysis**. To identify the m6A-binding proteins that are involved in mRNA decay, we collected cytosolic lysates from 0 to 2-h embryos of *Drosophila*, and performed a biotinylated RNA pull-down assay followed by the LC–MS/MS analysis. For obtaining cytosolic lysates, 1 g 0–2-h *w$^{1118}$* embryos were harvested and resuspended in 3 ml hypotonic buffer (10 mM Tris-HCl, pH7.5, 10 mM KCl, 1.5 mM MgCl$_2$, and protease inhibitors). The resuspended embryos were transferred to daunce and homogenized with tight stroke 50 times and centrifuged at 400*g* for 1 min at 4 °C. The supernatant was centrifuged at 15,800*g* for 15 min at 4 °C. The protein concentration of cytosolic lysates was examined using BCA protein Assay Kit (ThermoFisher, 23227). Biotinylated RNA pull-down assay was performed using the described method[17]. Briefly, 10 µl streptavidin sepharose high-performance beads (GE Healthcare, 17-5113-01) were used for each pull-down assay. Beads were first washed twice in 1 ml of RNA binding buffer (50 mM HEPES, pH 7.5, 150 mM NaCl, 0.5% NP40, 10 mM MgCl$_2$). To inactivate and remove RNases, we incubated the beads in 100 µl RNA binding buffer containing RNase inhibitor (0.8 U/µl) for 30 min on ice. After centrifugation and removal of RNasin–RNase complexes, beads were pre-blocked with yeast tRNA (50 µg/ml; Invitrogen, AM7119) in RNA binding buffer overnight at 4 °C. The pre-blocked beads were washed twice with RNA binding buffer and then incubated with 5 µg of biotinylated RNA probe diluted with RNA binding buffer to a final volume of 600 µl. Beads were incubated for 30 min at 4 °C in a rotation wheel to allow binding of biotinylated probes to the streptavidin beads. The beads were then washed once with 1 ml of RNA washing buffer (50 mM HEPES, pH 7.5, 250 mM NaCl, 0.5% NP-40 and 10 mM MgCl$_2$) and twice with protein incubation buffer (10 mM Tris-HCl, pH 7.5, 150 mM KCl, 1.5 mM MgCl$_2$, 0.1% NP-40, 0.5 mM DTT and protease inhibitors). The beads containing immobilized RNA were then incubated with 1 mg of cytosolic extract from 0–2-h embryos in a total volume of 600 µl protein incubation buffer for 2 h at 4 °C. After washing 3 times with protein incubation buffer, the beads were incubated at 95 °C for 10 min to elute the complexes.

For LC–MS/MS analysis, the eluted complexes were analyzed using 10% SDS-PAGE for a separation distance of 0.5–1 cm from stacking gel, then stained with coomassie brilliant blue minimally and cut for the following digest using trypsin (Promega, V5111). Tryptic peptides were separated with an Easy-nLC 1000 connected online to an Orbitrap Elite mass spectrometer (ThermoFisher). Peptides were separated with a 90 min total LC gradient (80 min acetonitrile gradient from 3 to 30%, and washed by 100% acetonitrile for 10 min) for data acquisition. For measurements on the Orbitrap Elite, the top 20 most abundant peptides were fragmented for every full scan with dynamic exclusion enabled and set to 30 s.

**Co-IP and LC–MS/MS analysis**. The 2–4-h embryos of indicated genotypes were harvested and the total lysates were prepared with lysis buffer (50 mM Tris-HCl, pH 7.4, 150 mM NaCl, 1% Triton X-100, 10% glycerol, 1 mM EDTA, and protease inhibitors). Samples of total lysates were subjected to immunoprecipitation (10 mg protein/sample) using the anti-*Drosophila* FMR1 antibody (1:100) or IgG overnight

at 4 °C. The mixtures were incubated with the Pierce protein A/G agarose for 4 h at 4 °C. Beads were then washed using lysis buffer for 1 h and incubated at 95 °C for 10 min to elute the complexes. For western blotting, samples were performed with corresponding primary antibodies by using standard protocols. For LC–MS/MS analysis, samples were separated using 10% SDS-PAGE and digested in-gel using Trypsin. Tryptic peptides were analyzed using the same parameters as described in the biotinylated RNA pull-down assay followed by the LC–MS/MS analysis.

**MaxQuant analysis**. We used label-free quantification for performing the relative quantification in MS-based biotinylated RNA pull-down and co-IP experiments. Briefly, all of their raw data of protein LC–MS/MS were analyzed using the MaxQuant software (v1.6.1.0). Tandem mass spectra were further searched against UniProt database containing 309323 sequences (uniprot_taxonomy_7215), released in March 2019 for protein identification. Mass tolerance was set to 4.5 ppm (precursor ions) and 20 ppm (fragment ions), the maximum missed tryptic cleavages was set to 2. Variable modifications were methionine oxidation and protein N-terminal acetylation. The fixed modification was cysteine carbamido-methylation. The minimum length of amino acids per peptide was 7. A target decoy search approach was applied for the peptide and protein identification with the default MaxQuant setting of 1% FDR. The post MaxQuant analysis included filtering the generated "protein groups.txt" table for contaminants. Proteins that were all identified in three replicates were used for the later analysis. We used the ratio of LFQ intensity (m6A vs. A or FMR1 antibody vs. IgG) to screen significantly changed proteins. Proteins with significant differential levels were identified by a one-tail *t*-test. For preferentially m6A interacting proteins, they were defined as those with at least twofold upregulated (*q*-value < 0.05) in m6A-modified probe samples. For specifically m6A interacting proteins, they were defined as those only appeared in m6A-modified probe samples. Similarly, proteins that were upregulated more than twofold (*q*-value < 0.05) in three biological replicates compared with control IgG were considered as FMR1-associated proteins.

**SPR measurements**. SPR experiments were performed using Biacore 8 K instrument (GE Healthcare). His-MBP tagged proteins were captured on sensor chip CM5 (2000–5000 response units). Biotin-A/m6A probes with increasing concentrations were injected into the protein surface for 1 min at a flow rate of 30 µl/min, dissociated for 1 min in running buffer (20 mM HEPES, pH 7.4, 150 mM KCl, 1.5 mM MgCl$_2$, and 0.1% NP40) at 25 °C. Equilibrium and kinetic constants were calculated by a global fit to the 1:1 Langmuir binding model (Biacore 8 K evaluation software).

**Quantitative real-time PCR analysis (q-RT-PCR)**. Embryos were collected with indicated stages and genotypes for RNA extraction. Total RNA was extracted using Trizol reagent and cDNA was generated with a FastQuant RT Kit (Tiangen, KR106). Actin5C was used as the constitutive control for the normalization of candidate gene expression. For RIP-q-RT-PCR, the RNAs were immunoprecipitated by anti-*Drosophila* FMR1 antibody or IgG. q-RT-PCR reactions were performed in triplicate on a CFX Connect Real-time System (Bio-Rad) using the UltraSYBR mixture (Cwbio, CW0957). The detailed information of primers is described in Supplementary Table 3.

**Poly(A)-tail (PAT) assay**. Total RNAs were isolated from embryos with indicated stages and genotypes using TRIzol reagent. The poly(A) tail length was tested with Poly(A) Tail-Length Assay Kit (ThermoFisher, 76455) following the manufacturer's instructions. PCR products were analyzed on a 2% agarose gel. The detailed information of primers is described in Supplementary Table 3.

**Embryo staining and fluorescence analysis**. For immunohistochemistry, embryos were fixed in fixation buffer (4% formaldehyde and 0.3% Tween-20 in PBS) for 30 min and washed in PBT (0.3% Tween-20 in PBS) for 15 min. Fixed samples were incubated with FMR1 (1:2000) or pAbp (1:2000) antibodies at 4 °C overnight and washed 3 times. Then the samples were incubated with secondary antibody at room temperature for 2 h, followed by washing for 3 times (10 min per time) in PBT. For tracing the GFP fluorescence in living embryos, the embryos at the indicated stages were collected and immediately examined under Nikon Eclipse TI microscopy. All the images were acquired using Nikon software on a Nikon Eclipse TI microscopy. The granule size was calculated with the analyzed particle tools in Image J 1.46r. Line profile analysis was used to analyze the correlation of FMR1 and pAbp proteins. Fluorescence intensity along the straight line of FMR1 and pAbp proteins was calculated with the plot profile tool in Image J 1.46r. The Pearson's correlation coefficient r values of two fluorescence signals were calculated with Excel.

**Phase separation assay**. Recombinant GFP-FMR1$^{FL}$ and GFP-FMR1$^{ΔLC}$ were expressed and purified from *Escherichia coli*. The phase separation was generated by mixing diluted GFP-FMR1$^{FL}$ or GFP-FMR1$^{ΔLC}$ protein (10 µM) with NaCl buffer (20 mM Tris-HCl, pH 7.5, with a dose gradient of NaCl) on a glass-bottom cell culture dish. For RNA-dependent droplet-formation experiments, proteins were incubated with Cy3-labeled A/m6A ssRNA or m6A probes in the buffer

(20 mM Tris-HCl, pH 7.5, 110 mM NaCl). Samples were immediately examined under Zeiss LSM 710 Meta confocal microscopy and images were captured following the indicated time points.

**Fluorescence recovery after photobleaching**. FMR1 droplets were photobleached and imaged with a 488 nm laser using Nikon Eclipse TI microscopy. At each time point, fluorescence intensity within the bleaching spot was divided by the intensity of a neighboring unbleached area of the same size to correct the changes. The ratio of recovery was analyzed from three independent biological replicates.

**SDD-AGE assay**. The indicated embryos were harvested and lysed in lysis buffer (50 mM Tris-HCl, pH 7.5, 50 mM NaCl, 0.5% Triton X-100, 5% glycerol with protease inhibitors) for 30 min, followed by centrifugation at the speed of 13,500$g$ at 4 °C. The supernatant was then diluted with loading buffer (0.5× TBE, 10% glycerol, 2% SDS, 0.0025% bromophenol blue) at room temperature for 30 min, and loaded onto a newly-prepared 1.5% agarose gel with 0.1% SDS. After electrophoresis in the running buffer (1× TBE and 0.1% SDS) for 40 min with a constant voltage of 100 V at 4 °C, the proteins were transferred to BioTrace NT nitrocellulose (PALL, 66485) for immunoblotting with indicated antibodies.

**RNA-seq data analysis**. We performed RNA-seq for wild-type, *mettl3–mettl14* double maternal mutant and *fmr1* maternal mutant embryo samples at the indicated stages using NovaSeq platform with 150 bp paired-end high-throughput sequencing. Low-quality reads and adapter sequences were trimmed for all raw reads using fastp software[55] with default parameters except for "-q 3 -u 50 -length required 150". The FastQC software (http://www.bioinformatics.babraham.ac.uk/projects/fastqc/) was used to confirm the high quality of the sequencing data. The clean reads were mapped to the December 2014 assembly of the *D. melanogaster* genome (UCSC version dm6, BDGP Release 6, unlocalized scaffolds excluded) with TopHat version 2.0[56] with parameter "--keep-fasta-order". The FPKM (per kilobase of coding exon per million fragments mapped) value for gene expression and differential expression analysis was carried out by cuffdiff from the cufflinks package (version 2.21)[57]. K-means clustering of the differentially expressed genes in the five developmental stages was evaluated using Cluster 3.0 (clustering method: K-means clustering, Distance metric: Euclidean distance)[58]. The clustered file was identified as (1) degraded maternal genes: if their original FPKM values >10 at 0–0.5-h stage and expression levels were consistently reduced thereafter; (2) stable maternal genes: if their original FPKM values >10 at 0–0.5-h stage and expression levels were relatively stable upon a time; (3) zygotic genes: if their original FPKM values <10 at 0–0.5-h stage and expression levels were induced after 2–3-h stage. The clustered gene groups with original FPKM values were imported to Excel and R for statistical processing and visualizing. The medians of FPKM values of each group were used to visualize expression data.

**MeRIP-m⁶A-Seq data analysis**. After performing RNA immunoprecipitation (RNA-IP) experiments using specific anti-m6A antibodies, all samples were sequenced on an Illumina HiSeq 4000 platform with paired-end 150 bp read length. For alignment of reads, adapter-trimming and low quality read filtering were firstly performed by the Cutadapt software (version 1.12)[59], then the remaining reads were aligned to *Drosophila* reference genome (UCSC version dm6, BDGP Release 6, unlocalized scaffolds excluded) by HISAT2 (version 2.1.0)[60] with default parameters. After discarding reads that were mapped to more than one genomic region, the uniquely mapped reads were processed to peak calling analysis. The call peak module in MACS2 (version 2.2.1)[61] was used to identify enrichment peaks with default parameters except for "-no model", and the corresponding input samples were serving as control. The two replicates of m6A peaks were intersected with each other by Bedtools package (version 2.25.0)[62] with the minimum overlap required as a fraction of them (80%). The overlapped peaks between two replicates were identified as high-confident m6A peaks. Differentially methylated sites on transcripts were identified by diffReps (version 1.55.3). These peaks identified were mapped to transcripts using homemade scripts. The motifs enriched in m6A peaks were detected using Find Individual Motif Occurrences (FIMO)[63] on the peak summits 200 bp of sequences.

We normalized m6A changes between conditions by using ngs.plot Two-step Normalization. In brief, we calculated the number of reads of m6A–RIP and corresponding input in each sample that mapped to each window or region by using ngs.plot software (v2.61) with two-step normalization. Firstly, the number of reads in the m6A–RIP and input were normalized by reading counts per million mapped reads, and then the normalized m6A–RIP read number was divided by the read number of the corresponding input as enrichment score of that window or region. Significant differences of normalized m6A coverage between the conditions were determined with a 95% confidence interval and Wilcoxon rank-sum test of the R stats library package. In order to get the 95% confidence interval of enrichment score for each window or region, we sampled the peaks 100 times by using the -S parameter of ngs.plot software, and the resulting enrichment scores were subjected to seaborn (a Python data visualization library) to plot coverage with 95% confidence interval for the specific regions in the study. To evaluate the statistical significance of m6A

coverage, we divided the range (~±1000 bp around the peak summit) into 60 bins. Then the statistical significance was estimated by comparing the values representing the coverage of m6A after normalization in the ten bins around the summit (~±300 bp) using the Wilcoxon rank-sum test.

**Data-dependent acquisition (DDA) and data analysis**. Embryos from different stages were lysed using a buffer containing 8 M urea, 50 mM Tris-HCl (pH 8.0), and protease inhibitor cocktail, and the tissue was centrifuged (12,000$g$, 15 min). To build a spectral library for following DIA analysis, we firstly used an equal amount of samples applied in DIA to generate a data-dependent acquisition (DDA) spectral library. The samples were mixed and then subjected to SDS-PAGE gel, which was further cut into 10 pieces from high to low molecular weight. The proteins in each piece of gel were digestion by trypsin overnight at 37 °C, and then resuspended in a solution of 0.1% formic acid in water containing proportionally added iRT reagent and then further analyzed by Orbitrap Exploris 480 MS system. Proteome Discoverer ver. 2.4 (ThermoFisher) was used to search the above 10 groups of DDA mass spectrometry results. Peptide/protein entries identified were used to build the final DDA spectral libraries using Skyline software (Skyline 21. 1. 0. 146) for the following DIA data analysis[64]. The *Drosophila* reference proteome was downloaded from UniProt on January 14, 2021.

**DIA quantitative proteomics and analysis**. Embryos from different stages were lysed using a buffer containing 8 M urea, 50 mM Tris-HCl (pH 8.0), and protease inhibitor cocktail, and the tissue was centrifuged (12,000$g$, 15 min). The supernatant was harvested and digested by trypsin (1:80 weight/weight (w/w)) in the solution overnight at 37 °C. The peptides were desalted with a C18 column. The concentrations of the peptides obtained after trypsin digestion were determined using nanodrop. Digested samples that were added with a proportion of iRT reagents (Biognosys, Ki-3002-2) were analyzed on an Orbitrap Exploris 480 mass spectrometry system equipped with an Easy nLC 1200 ultra-high pressure liquid chromatography system (ThermoFisher).

The DIA data was analyzed using Skyline software (Skyline 21. 1. 0. 146), which used iRT peptides to calibrate retention time. The default parameters were used for the DIA data analysis, in which the protein and peptide FDRs were set to less than 5%. The protein expression levels of *fmr1* maternal mutant groups were compared with those of the wild-type group, and the ratio between groups was regarded as the expression change of a protein or peptide, which was also the basis for further data analysis. Quantified proteins with at least an expression change value >1.5-fold, $q$-value < 0.05 were considered to have significant expression changes.

**Reporting summary**. Further information on research design is available in the Nature Research Reporting Summary linked to this article.

## Data availability

The data supporting the findings of this study are available from the corresponding authors upon reasonable request. The raw sequencing data generated in this study have been deposited in the NCBI's Gene Expression Omnibus and are accessible through GEO Series accession number GSE143821. The mass spectrometry proteomics data generated in this study have been deposited to the ProteomeXchange Consortium via the iProX partner repository[65] with the dataset identifier PXD026356. Source data are provided with this paper.

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

## Acknowledgements

This work is supported by the Basic Science Center Program of NFSC (31988101), the National Key R&D Program of China (2017YFA0506800, 2018YFC1003300, and 2019YFA0802100), and NFSC projects (32170818).

## Author contributions

G.Z., Q.S., and D.C. designed the experiments; G.Z., Y.X., Y.Z., W. Z., Y.W., and Y.G. performed the experiments; Xi.W., G.Z., L.W., Y.C., Xu.W., Q.S., and D.C. analyzed the data; G.Z., Q.S., and D.C. wrote the paper. All authors provided intellectual input, vetted, and approved the final paper.

## Competing interests

The authors declare no competing interests.
