## [Peer Review File · Nature Communications]

Title: Dynamic FMR1 granule phase switch instructed by m6A modification contributes to maternal RNA decayREVIEWER COMMENTS

Reviewer #1 (Remarks to the Author):

The authors found that loss of N6-methyladenosine(m6A) in Drosophila embryos resulted in deficiency of hatching rate and a delay of maternal mRNA clearance. They showed that loss of FMR1, a m6A reader protein, could phenocopy loss of the methyltransferase. FMR1 preferentially binds “AGACU” motif and is engaged in granule formation at 2-3 h of development. m6A-containing oligos could enhance the propensity of purified FMR1 to form granules. The authors proposed that FMR1 in condensates could enhance its propensity to recruit mRNA decay machinery and the decay of maternal mRNA in turn, lead to the disassembly of FMRP-containing RNPs.

This work provides evidence that FMR1 is involved in maternal mRNA decay during Drosophila embryo development. It also provides a possible molecular mechanism of how FMR1 phase separation affects its function. The conclusions in this manuscript are well supported by properly designed experiments. I have a few concerns below:

Major concerns:

1) The presentation of Fig. 5a and Fig. 5b is strange. Group 3 RNA was shown to be enriched of m6A, however, the authors defined different groups of RNA with protein level/RNA level, which is an estimate of translation efficiency, instead of stability. Group 2 RNA are actually translation efficiency upregulated RNAs, while group 3 RNA are translationally repressed RNA. The authors should group RNA based on m6A modification status or more preferentially, FMR1 binding targets upon FMR1 deletion to show FMR1 controls their degradation. It is also weird that there were almost no downregulated mRNA in Fig. 5a, exhibiting very different pattern from Fig. S4b.

2) The authors did not rule out the possibility that FMR1 regulates translation rates and thus, affects coupled RNA dedenylation. In Fig S10C, the majority of FMR1-interacting proteins are ribosomal proteins. The authors should use protein MS-Spec and RNA-Seq data obtained from Fmr1^{-/-}KO and control strain to perform more careful analysis to dissect effects of FMR1 on translation and RNA degradation. RNA lifetime assay should help elucidate the mechanism.

Minor concerns:

- 1) Fig. 1f, other works showed that the YTHDF protein is m6A reader in drosophila, why doesn't it bind m6A here?
- 2) Fig. S1a, how will Fmr1 deletion affect mRNA m6A level during development?

Reviewer #2 (Remarks to the Author):

In this manuscript by Zhang et al., the authors present convincing evidence reinforcing other studies that FMR1 can bind to m6A methylated GGACU sequences. They also demonstrate a link between FMR1 and programmed decay of maternal mRNAs. The paper is an interesting addition to the literature on the role of FMR1 in development, adding an additional layer onto the finding that FMR1 is involved in regulation of translation.

Major Comments:

The authors present results suggesting that the canonical m6A RNA binders Ythdf and Ythdfc do not bind to their m6A-modified RNA probes. Since this runs counter to the expected results, the authors should present the evidence that their Ythdf antibody does in fact recognize Ythdf to make this more convincing.

Discussion of the DIA quantitative proteomic results is also difficult to follow. There are likely additional explanations for the data besides the two considered by the authors. The existing explanation is not satisfactory.

The discussion of selective maternal vs zygotic RNA decay is also somewhat unclear.

In addition, the roles of each of the potential RNA-binding elements in FMR1 was not fully addressed or discussed. KH0, KH1 and KH2 are all possibly able to bind RNA, with Agenet1 (2-49), Agenet2 (63-113), KH0 (126-202), KH1 (216-280), KH2 (280-404) defined in this paper: Myrick, Hashimoto, Cheng, Warren. Human FMRP contains an integral tandem Agenet (Tudor) and KH motif in the amino terminal domain. Hum Mol Genet. 2015 Mar 15; 24(6): 1733–1740. doi: 10.1093/hmg/ddu586. The RGG region can also bind RNA, and the different specificity of this region was not discussed.

In addition to addressing these significant issues, the authors should address these specific comments:

Pg. 3. “that FMR1 could bind the m6A modified target mRNAs...” Add citation: Arguello et al. (PMID 29140688), who have demonstrated that FMR1 (from HELA cells) can bind to methylated GGACU sequences.

Pg. 8. “but Ythdf was not detectable in these complexes”. How was the Ythdf antibody specificity verified? There is a band on the western for the input, but how did the authors verify that this is Ythdf? It seems odd that Ythdf was not detectable in these complexes. Verification of the specificities of the antibodies, which the authors generated, should be documented in the supplementary information.

Figure S6e,f – These are very unusual SPR curves. Consider removing these data, since the interpretation could be very tricky.

Pg. 13. “indicating that the V311 residue is critical for preferential binding of FMR1 to RNAs containing the m6A-modified AGACU motif.” and Pg. 24. “binding to the specific m6A-modified motif requires the presence of a single specific residue, V311” These statements are incorrect. It could be that a lysine in this position strengthens the overall affinity of FMR1 for the RNA probe, such that methylation of the adenine residue is no longer required for high affinity binding. The V311 could be irrelevant and is certainly not required as demonstrated by the excellent binding achieved when V311 is not present (in the mutant).

Pg. 16. The authors should provide a Coomassie stained gel of the purified GFP-FMR1 and GFP-FMR1deltaLC in the supplementary material to demonstrate the level of purity and describe the purification protocol.

Figure 4a, 4f The merged images look like the GFP-FMR1 images. To fix this, perhaps change the coloring of the merged image or exclude the merged images altogether, since the overlap is obvious. Also consider enhancing the contrast in the Cy3 images to make the red a little easier to see.

Pg. 17. “suggesting that binding to m6A-modified RNA significantly changes the FMR1 condensate solubility” is not correct. Change "solubility" to "fluidity" or "dynamics".

Pg. 18. “presence of m6A-marked RNA is critical for dynamics” is ambiguous. Does this mean that it is critical for the formation of FMR1 granules or that the granules themselves are dynamic and fluid? Consider changing to “presence of m6A-marked RNA increases the size of FMR granules and enhances the partitioning of non-methylated RNA into the granules”.

Pg. 18. “in regulating the dynamics of FMR1 granules” should be “in enhancing FMR1 granule size and partitioning of RNA molecules into the granules”.

Pg. 18 and 19. The logic behind the interpretation of these results is difficult to follow. There are more possibilities than the two options (1. FMR1 directly regulates through RNA decay and 2. aberrant stability of mRNAs induced by loss of maternal FMR1 is a “byproduct” of translational repression) listed. Is there a competition between RNA decay and translation repression modulating the final protein levels? How can the data for Group 2 (mRNA no change, protein up) be explained? Is there a role for nuclear export of m6A labelled mRNA involving FMRP (Edens et al. reference) that might help explain these results?

Pg. 19 “were significantly correlated with m6A modification (Fig. 5b).” What specifically is being correlating here? That m6A modifications are enriched within group 3 relative to group 2? Or to group 2 and group1? A visual inspection of the plots suggests that the phrasing here is incorrect.

Pg. 21 “little is known about how maternal RNA decay is triggered and how maternal RNAs (but not zygotic RNAs)” The paper does not really clearly explain preferential degradation of maternal RNA over zygotic RNA. Is m6A specifically enriched on maternal RNA or is there at least a delay in the methylation

of the zygotic RNA?

Does FMR1 recognize m6A modified RNA on its own or does it do this with Ythdf (see BiorXiv doi: <https://doi.org/10.1101/2020.03.04.976886> and reference this.) This paper should be cited as it discusses the role of Ythdf and ythdc1 as major m6A binders in *Drosophila*. Nat. Commun. 2021 Mar 5;12(1):1458.doi: 10.1038/s41467-021-21537-1. A neural m6A/Ythdf pathway is required for learning and memory in *Drosophila*.

This additional reference should be included:

Zhang, F., Kang, Y., Wang, M., Li, et al. Fragile X mental retardation protein modulates the stability of its m6A-marked messenger RNA targets. Hum. Mol. Genetics., 27, 22, 2018.

Reviewer #3 (Remarks to the Author):

The authors test the role of methylated mRNAs and the RNA binding protein Fmr1 in embryonic development. They focus on understanding the embryonic degradation of maternally loaded mRNAs, which is a necessary step in the switch from maternal to zygotic control of embryogenesis. The authors claim that they have discovered an essential role for m6A marks in the degradation of maternal mRNAs, through the binding of methylated mRNAs by Fmr1. Furthermore, Fmr1 appears to preferentially bind to m6A-marked mRNAs, and forms particles dynamically during embryogenesis, which may indicate a dynamic function such as mRNA turnover or another function not explored by the authors.

I have major concerns about this interpretation of the data in the manuscript, which in my opinion undermines the conclusions made by the authors:

(1) Are m6A and Fmr1 required for maternal mRNA degradation? The authors own evidence indicates that there is unlikely to be a major/direct role for either in maternal mRNA degradation.

About 1/3 of maternal mRNAs are marked for degradation during the first ~3 hours of *Drosophila* embryogenesis (De Renzis et al. 2007), a process which requires RNA decay factors such as Smaug (Benoit et al. 2009). While ~25% of degraded maternal transcripts are cleared through strictly maternal machinery, ~75% are degraded through zygotic machinery or by both maternal/zygotic mechanisms (Thomsen et al. 2010, Tadros et al. 2007, De Renzis et al. 2007). Surprisingly, the authors observe no effect on mRNA turnover at the 2-3 hour timepoint in which most maternal mRNAs are turned over, either in mutants with reduced m6A (Figure 1D) or in Fmr1 mutants (Figure 2C).

The authors only observe differences in maternal mRNA levels several hours after zygotic genome activation is normally induced. Any perturbations that reduce zygotic transcription would have the effect in RNA sequencing data of making maternal transcripts appear more abundant. Indeed a large fraction (~25%) of Fmr1 null mutant embryos fail to properly cellularize, an event intimately tied to

zygotic genome activation (Monzo et al. 2006). There is no attempt by the authors to show that Fmr1 or Mettl3-14 mutants lead to specific defects in RNA turnover rather than general defects in the developmental progression of embryos. The authors may have come to the same conclusions by examining nonspecific mutants blocking normal developmental progression.

(2) Hundreds of mRNAs are translationally misregulated in Fmr1-deficient oocytes prior to embryogenesis (Greenblatt et al. 2018). How can the authors exclude contributions of altered levels of one or more of the hundreds of Fmr1 targets defects? It is important to note that the RNA decay factor Not1 is one such target of Fmr1 translationally upregulated by in oocytes.

(3) Many studies have been performed on Fmr1-bound mRNAs. There are no specific motifs that have emerged that are consistently highly enriched among Fmr1 target mRNAs as compared to non-targets. Multiple groups have found rather that mRNA length appears to be the major determinant in Fmr1 binding (Li et al. 2020, Sawicka et al. 2019). Is there any correlation with mRNA length and mRNA stabilization in Fmr1 or Mettl3-14 mutants at the 5-6 hour timepoint? In addition, is there any correlation between oocyte Fmr1 targets and those stabilized in Fmr1 mutant 5-6 hour embryos? If these relationships exist it would strengthen the authors' claim that FMRP is playing a direct role in the process they are studying.

(4) If FMRP granule formation is important for RNA decay, why is it occurring at a timepoint in which there is no apparent difference in RNA levels in WT vs. Fmr1 mutants? If these granules are playing a direct role, can the authors find enrichment for any mRNA supposedly targeted by Fmr1 for degradation in these granules?

(5) In comparing Group 2 and Group 3 genes that show similar profiles in Fmr1 mutant and mutants lacking m6A (Fig. 5), the authors make a strange finding that mRNAs that are increased in their sequencing data in 5-6 hour Fmr1 mutants and mutants lacking m6A do not show an increase in protein abundance, whereas only mRNAs that are unchanged show an increase in protein abundance. This appears to be inconsistent with known RNA turnover mechanisms in the early embryo. mRNA decay induced by known RNA degradation factors such as Smaug and Me31B is intimately associated with translational repression. Why would the Fmr1/m6A-dependent degradation system operate differently from known RNA degradation pathways?

(7) As far as I can tell there is no attempt to show that the mRNAs that are stabilized in m6A-deficient or Fmr1 mutant embryos are actually themselves methylated. This is an important point, since it would address whether the effects of methylation are direct or indirect.

(8) Finally, the authors should show using qPCR, Northern Blot, or RNA seq with proper normalization controls, that the effects they observe are due to increased levels of maternal mRNAs at the 5-6 hour timepoints rather than decreased zygotic gene expression leading to an RNA-seq normalization artifact.

Reviewer #4 (Remarks to the Author):

Figure 1e/S3d/e corresponding to RNA pull down coupled to LC-MS/MS to identify m6a interacting proteins

In Figure 1e the author show the schematic representation of RNA pull down coupled to LC-MS/MS to determine proteins that preferentially and/or specifically bind to m6a RNA. This is a nice experiment but for unclear reasons the authors fail to show any results that they have obtained.

- What kind of quantitative approach was used to determine proteins that preferentially bind to m6a RNA over unmodified RNA (and the other way around). Without a quantitative approach the authors cannot make any claims about preferential binding.
- How many biological replicates were performed?
- The authors need to use a quantitative approach of their choice and then use a statistical approach based on FDR and fold change to determine proteins preferentially binding to m6A RNA. I suggest showing the data in a volcano plot as usually done for quantitative proteomics data. This should be part of Figure 1.
- Were any expected proteins identified that are known to bind to m6a RNA (these ones should also then be indicated in the volcano plot)
- A table with all quantified protein groups with quantified fold changes and FDR should be provided as supplementary dataset.
- All raw proteomics data should be uploaded to a public repository such as Pride (<https://www.ebi.ac.uk/pride/>) (from Figures 1, S3, 5 and S10)
- Can the authors provide an explanation in the methods what they mean by cytosolic fraction of proteins?
- Can the authors provide explanation why known m6A binder YTHDF does not bind the m6a probe in western blot in Figure 1e?
- Which Drosophila dataset was used for MaqQuant analysis (i.e. number of proteins and release date)

Figure S3d/e corresponding to FMR1 co-IP coupled to LC-MS/MS

Here even less information is provided compared to Figure 1. Similar questions should be addressed:

- What kind of quantitative approach was used to determine proteins that preferentially bind to FMR1 compared to IgG. Without a quantitative approach the authors cannot make any claims about proteins interacting to FMR1.
- The authors need to use a quantitative approach of their choice and then use a statistical approach based on FDR and fold change to determine proteins co-immunoprecipitating FMR1. I suggest showing the data in a volcano plot as usually done for quantitative proteomics data.
- A table with all quantified protein groups with quantified fold changes and FDR should be provided as supplementary dataset.
- In methods section, it is not stated which antibody and what protein amount was used for co-IP and if total cellular lysates or fractionated proteins were used.

Figure 5A/B corresponding to protein levels in wild-type and fmr1 mutant embryos

Here, I cannot find any information about the total proteome analysis in wild-type and fmr1 mutant embryos either in the main text or methods section. The authors need to provide information about this. Did the authors measured total proteomes of wild-type and fmr1 mutant embryos and again how was this done and what kind of quantitative approach was used. I could have missed that because figures seem to have a random order in the text.

Figure S10 corresponding to DIA-MS analysis of embryo-derived proteins

Why was here DIA analysis used? The rationale why DIA comes into play here is not clear in the manuscript text at all. Which specific proteins were authors interested to monitor? I realize now that this is probably the data used also in Figure 5. The orders of figures need to be re-organized and it has to be clear in the main text what is shown in Figure 5.

Other points

What version of the Skyline software was used for the analysis?

The figures do not flow correctly in the manuscript so it is very difficult to read the manuscript and identify the correct figure.

I noticed that in some figures authors use non-corrected p-values in RNA-seq analysis instead of FDR/q-value.

In the methods title DIA-MS, authors also explain the DDA-MS. This has to be changed and DDA has to be included in the other paragraph

Responses to Reviewer #1's comments:

The authors found that loss of N6-methyladenosine (m6A) in *Drosophila* embryos resulted in deficiency of hatching rate and a delay of maternal mRNA clearance. They showed that loss of FMR1, a m6A reader protein, could phenocopy loss of the methyltransferase. FMR1 preferentially binds “AGACU” motif and is engaged in granule formation at 2-3 h of development. m6A-containing oligos could enhance the propensity of purified FMR1 to form granules. The authors proposed that FMR1 in condensates could enhance its propensity to recruit mRNA decay machinery and the decay of maternal mRNA in turn, lead to the disassembly of FMRP-containing RNPs.

This work provides evidence that FMR1 is involved in maternal mRNA decay during *Drosophila* embryo development. It also provides a possible molecular mechanism of how FMR1 phase separation affects its function. The conclusions in this manuscript are well supported by properly designed experiments. I have a few concerns below:

Response: We thank the reviewer for positively commenting on our manuscript.

Major concerns:

1) The presentation of Fig. 5a and Fig. 5b is strange. Group 3 RNA was shown to be enriched of m6A, however, the authors defined different groups of RNA with protein level/RNA level, which is an estimate of translation efficiency, instead of stability. Group 2 RNA are actually translation efficiency upregulated RNAs, while group 3 RNA are translationally repressed RNA. The authors should group RNA based on m6A modification status or more preferentially, FMR1 binding targets upon FMR1 deletion to show FMR1 controls their degradation. It is also weird that there were almost no downregulated mRNA in Fig. 5a, exhibiting very different pattern from Fig. S4b.

Response: We apologized for the confusion of data presentation in Fig.5a and Fig.5b. Actually, in the Fig. 5a of the original version, we indeed used protein MS-Spec datasets and RNA-Seq datasets generated from *fmr1* maternal mutant embryos and wild-type embryos at the 5-6 hour stage to analyze integrative changes of gene expression at both mRNA and protein levels. As suggested by Reviewer#1 in the no. 2 comment, the

purpose of Fig.5a in the original version is for asking whether FMR1 has a role in regulating translation repression (or controlling translation rates), maternal RNA degradation or both.

As Reviewer #1 pointed out, there was a possibility that FMR1 regulates translation repression, thus indirectly affecting decay of a portion of coupled maternal mRNAs. If it were a case in the early embryos, one would have also seen aberrant upregulation of protein products corresponding to aberrant upregulated mRNAs in *fmr1* maternal mutant, when compared to wild-type control. To address this important issue, we analyzed the proteomic datasets together with RNA-seq datasets, and identified the 4814 hits (**Fig. S10b in the revision**). Because FMR1 is a well-known translation repressor and potentially regulates RNA decay, we then focused on analyzing the up-regulated genes and no significant changed genes at either mRNA or protein levels. As shown in **Fig. 5a and Fig. S10c in the revision**, we classed these genes into three major groups: genes in the group 1 showed no significant change at both protein and mRNA levels; genes in the group 2 displayed increase at protein levels, but no significant change in levels of mRNAs; and genes in the group 3 showed increase at mRNA levels, but no significant change in levels of proteins. Interestingly, we found that majority of targets with aberrant upregulation at mRNA levels showed decrease or no change in levels of proteins (group 2), suggesting that FMR1 has a role in regulating maternal mRNA decay. Additionally, we also found that genes in the group 3 (but not group 2) were significantly correlated with m6A modification (**Fig. 5b in the revision**). Collectively, our findings suggest that in addition to translation repression, FMR1 indeed plays a role in maternal mRNA decay through m6A modification.

Regarding the relationship between Fig. 5a and Fig. S4b of the original manuscript, in the previous version, we attempted to highlight that FMR1 regulates gene expression through both translational repression and RNA decay in early embryos, and thus focused on presenting the data of group 2 and group 3. According to Reviewer #1's suggestion, we have revised the **Fig. 5a and b**.

2) The authors did not rule out the possibility that FMR1 regulates translation rates and thus, affects coupled RNA dedenylation. In Fig S10C, the majority of FMR1-interacting proteins are ribosomal proteins. The authors should use protein MS-Spec and RNA-Seq data obtained from Fmr1-KO and control strain to perform more careful analysis to dissect effects of FMR1 on translation and RNA degradation. RNA lifetime assay should help elucidate the mechanism.

Response: We thank Reviewer #1 for the suggestion. We believe that we have answered the question in the response to the no. 1 comment above.

Minor concerns:

1) Fig. 1f, other works showed that the YTHDF protein is m6A reader in drosophila, why doesn't it bind m6A here?

Response: We have performed extensive biochemical assays to test whether *Drosophila* Ythdf (dYthdf) binds m6A-modified RNA. 1) the gel-shift assays showed that dYthdf has no binding affinity to multiple m6A modified RNA probes (**the Response Figure 1a, b**). Of note, in the gel-shift experiments, we used the human Ythdf2 (hYthdf2) as a control, we found that hYthdf2 exhibited strong binding affinities to the probes in the same experimental conditions (**the Response Figure 1a, b**). 2) The SPR analysis again showed that dYthdf has no binding affinity to the m6A modified RNA probes (**the Response Figure 1c-f**). A recent study reported that probe containing "AAAmCU" probe could bind the dYthdf protein¹. We used both the gel-shift and SPR assay to test this possibility, and found that hYthdf2 could bind the AAAmCU, but dYthdf did not (**the Response Figure 1g-k**). Our results strongly suggest that in contrast to hYthdf2, the dYthdf has no apparent binding affinity to m6A-modified RNAs *in vitro*. Additionally, we also found that it is dispensable for *Drosophila* early embryonic development.

Response Figure 1. a, Coomassie brilliant blue staining of purified proteins as indicated. **b**, EMSAs showing the binding capability of hYthdf2 and dYthdf to RNA probes containing “GGACU”, “AGACU”, “GAACU” with or without m6A modification. **c** and **d**, SPR assays showing binding affinity of hYthdf2 protein with RNA probes containing “GGACU”, “AGACU”, “GAACU” with unmodified RNA probes (**c**) or with m6A-modified probes (**d**). **e** and **f**, SPR assays showing binding affinity of dYthdf protein with RNA probes containing “GGACU”, “AGACU”, “GAACU” with unmodified RNA probes (**e**) or with m6A-modified probes (**f**). **g**, EMSAs showing the binding capability of hYthdf2 and dYthdf to RNA probes containing “AAACU” (9x) with or without m6A modification. **h** and **i**, SPR assays showing binding affinity of hYthdf2 protein with RNA probes containing “AAACU” (9x) with unmodified RNA probes (**h**) or with m6A-modified probes (**i**). **j** and **k**, SPR assays showing binding affinity of dYthdf protein with RNA probes containing “AAACU” (9x) with unmodified RNA probes (**j**) or with m6A-modified probes (**k**).

2) Fig. S1a, how will FMR1 deletion affect mRNA m6A level during development?

Response: We thank the reviewer's comment. Based on our results, we propose that FMR1 only bind a portion of m6A-modified mRNAs with high affinity in a sequence-dependent manner. Through binding, these m6A-modified mRNAs likely act as "ligands" to promote condensation of FMR1 granules in the 2-3 hour stage, and the condensed granules then recruit other components and unmodified mRNA for further degradation in the later (5-6 hour) stage. In order to answer the reviewer's question, we have used the anti-m6A antibody to purify m6A-modified mRNA in *fmr1* maternal mutant and wild-type embryos. We then used the immunoprecipitants to perform the q-RT-PCR assays to test effects of FMR1 on m6A levels of target mRNAs. As shown in the Response Figure 2, relative levels of m6A on the tested transcripts containing "AGACU" in *fmr1* maternal mutant embryos appeared to increase when compared to that in wild-type embryos at the 5-6 hour stage.

Response Figure 2. q-RT-PCR experiments were performed to confirm the m6A levels of target mRNAs when *fmr1* maternal mutant samples were compared with that of wild-type at the 5-6 hour stage by using m6A antibody. In this assay, the corresponding target mRNAs IPed by IgG were used for normalization.

Responses to Reviewer #2's comments:

In this manuscript by Zhang et al., the authors present convincing evidence reinforcing other studies that FMR1 can bind to m6A methylated GGACU sequences. They also demonstrate a link between FMR1 and programmed decay of maternal mRNAs. The paper is an interesting addition to the literature on the role of FMR1 in development, adding an additional layer onto the finding that FMR1 is involved in regulation of

translation.

Response: We thank the reviewer for positively commenting on our manuscript.

Major Comments:

1. The authors present results suggesting that the canonical m6A RNA binders Ythdf and Ythdfc do not bind to their m6A-modified RNA probes. Since this runs counter to the expected results, the authors should present the evidence that their Ythdf antibody does in fact recognize Ythdf to make this more convincing.

Response: In response to the reviewer's comment, we have used lysates isolated from wild-type, *ythdf* maternal mutant and *ythdf* maternal overexpression embryos at the 0-1 hour stage to perform western blot assays. As shown in **Fig. S3e in the revision**, we have verified that the anti-dYthdf antibody specifically recognizes the *Drosophila* Ythdf protein.

Discussion of the DIA quantitative proteomic results is also difficult to follow. There are likely additional explanations for the data besides the two considered by the authors. The existing explanation is not satisfactory.

Response: Reviewer #1 made the similar comment. In the revision, we have revised the main text and figure legend of Fig. 5a and 5b, and made detailed responses to the no.1 major concern of Reviewer #1.

The discussion of selective maternal vs zygotic RNA decay is also somewhat unclear.

Response: In *Drosophila* embryos, the event of maternal-to-zygotic transition occurs from the 0.75 hour to the 5-6 hour stage. During this timing window, maternal RNAs undergo a large-scale degradation². Based on this, we collected samples within this developmental window. RNA binding proteins selectively bind target RNAs to form RNP complexes/granules that regulate their function and fate. In this study, we provided several lines of evidence to support that FMR1 acts in conjugation with m6A modification to regulate a portion of maternal mRNA degradation. First, m6A modification is required for dynamics of FMR1 RNP granules in early embryos, and

Mettl3/14 genetically interacts with FMR1. Second, RNA-seq analysis suggest that FMR1 and Mettl3/14 share a common set of maternal mRNA targets. Third, FMR1 binds a portion of m6A-modified mRNAs with high affinity in a sequence-specific manner. Based on these observations, we argue that FMR1-bound, m6A-modified mRNAs may serve as an important regulator to promote condensation of FMR1 granules in the 2-3 hour stage, and the condensed granules then recruit other components and unmodified mRNA for further degradation at the later (5-6 hour) stage.

By analyzing the datasets between wild-type and *mettl3-mettl14* double maternal mutant embryos, we found no significant up-regulation of the zygotic transcripts in *mettl3-mettl14* double maternal mutant embryos (**the Response Figure 3**). Taken together, we argue that maternal mRNAs are major targets for FMR1 and m6A.

Response Figure 3. No significant up-regulation of the zygotic transcripts in *mettl3 - mettl14* double maternal mutant embryos.

In addition, the roles of each of the potential RNA-binding elements in FMR1 was not fully addressed or discussed. KH0, KH1 and KH2 are all possibly able to bind RNA, with Agenet1 (2-49), Agenet2 (63-113), KH0 (126-202), KH1 (216-280), KH2 (280-404) defined in this paper: Myrick, Hashimoto, Cheng, Warren. Human FMRP contains an integral tandem Agenet (Tudor) and KH motif in the amino terminal domain. Hum Mol Genet. 2015 Mar 15; 24(6): 1733–1740. doi: 10.1093/hmg/ddu586. The RGG region can also bind RNA, and the different specificity of this region was not discussed.

Response: We thank the reviewer for the excellent comment. We agree with Reviewer #2's point that other domains in FMR1 play roles in binding RNA. However, given that FMR1 acts in conjugation with m6A modification to regulate a portion of maternal

mRNA degradation, in this study we focused on identifying the specific domain that is required for FMR1 to interact with m6A-modified RNA. As shown in **Fig. S6a, b in the revision**, Agenet domains, KH0 and RGG are not required for preferential binding of FMR1 to the m6A-modified RNA probes.

In addition to addressing these significant issues, the authors should address these specific comments:

Pg. 3. “that FMR1 could bind the m6A modified target mRNAs...” Add citation: Arguello et al. (PMID 29140688), who have demonstrated that FMR1 (from HELA cells) can bind to methylated GGACU sequences.

Response: We have included the paper in the reference list.

Pg. 8. “but Ythdf was not detectable in these complexes”. How was the Ythdf antibody specificity verified? There is a band on the western for the input, but how did the authors verify that this is Ythdf? It seems odd that Ythdf was not detectable in these complexes. Verification of the specificities of the antibodies, which the authors generated, should be documented in the supplementary information.

Response: We have used lysates isolated from wild-type, *ythdf* maternal mutant and *ythdf* maternal overexpression embryos at the 0-1 hour stage to perform western blot assays. We have verified that the anti-dYthdf antibody specifically recognizes the *Drosophila* Ythdf protein, and included the new results in the revised manuscript (**Fig. S3e in the revision**).

Figure S6e, f – These are very unusual SPR curves. Consider removing these data, since the interpretation could be very tricky.

Response: We have removed these data according to the reviewer’s suggestion in the revision.

Pg. 13. “indicating that the V311 residue is critical for preferential binding of FMR1 to RNAs containing the m6A-modified AGACU motif.” and Pg. 24. “binding to the

specific m6A-modified motif requires the presence of a single specific residue, V311” These statements are incorrect. It could be that a lysine in this position strengthens the overall affinity of FMR1 for the RNA probe, such that methylation of the adenine residue is no longer required for high affinity binding. The V311 could be irrelevant and is certainly not required as demonstrated by the excellent binding achieved when V311 is not present (in the mutant).

Response: It is a good point. We agree with the reviewer’s comment that a lysine may change the binding affinity between FMR1 and RNA probes, since the lysine is thought to be an important residue for protein modification. However, in our *in vitro* “gel-shift” assay, we used bacterial purified protein to test whether FMR1 interacts with the m6A-modified RNA probes or unmodified probes. The results excluded the possibility that FMR1 binds RNA through protein modification (e.g., ubiquitination). To strengthen our conclusion, we generated a new mutant of FMR1, in which V311 was changed to R. By performing “gel-shift” assays, we found that like the mutation of V311 to K, the mutation of V311 to R also damaged preferential binding of FMR1 to RNAs containing the m6A-modified AGACU motif (**the Response Figure 4**).

Response Figure 4. **a**, Coomassie brilliant blue staining of purified proteins. **b**, EMSAs were performed to detect the binding affinities of FMR1 proteins carrying indicated site mutation to RNA probes with or without m6A modification. The mutation of V311 to R damaged preferential binding of FMR1 to the m6A-modified RNA.

Pg. 16. The authors should provide a Coomassie stained gel of the purified GFP-FMR1 and GFP-FMR1deltaLC in the supplementary material to demonstrate the level of purity and describe the purification protocol.

Response: As the reviewer requested, we have confirmed purity of the purified proteins,

and included the new results in the revised manuscript (**Fig. S9a in the revision**).

Figure 4a, 4f The merged images look like the GFP-FMR1 images. To fix this, perhaps change the coloring of the merged image or exclude the merged images altogether, since the overlap is obvious. Also consider enhancing the contrast in the Cy3 images to make the red a little easier to see.

Response: As the reviewer suggested, we have excluded the merged images in the revised manuscript (**Fig. 4a, f in the revision**).

Pg. 17. “suggesting that binding to m6A-modified RNA significantly changes the FMR1 condensate solubility” is not correct. Change "solubility" to "fluidity" or "dynamics".

Response: Thanks for the comment. We have changed “solubility” to “fluidity” in the revised manuscript.

Pg. 18. “presence of m6A-marked RNA is critical for dynamics” is ambiguous. Does this mean that it is critical for the formation of FMR1 granules or that the granules themselves are dynamic and fluid? Consider changing to “presence of m6A-marked RNA increases the size of FMR granules and enhances the partitioning of non-methylated RNA into the granules”.

Response: Thanks for the comment. We have made a change in the revised manuscript, accordingly.

Pg. 18. “in regulating the dynamics of FMR1 granules” should be “in enhancing FMR1 granule size and partitioning of RNA molecules into the granules”.

Response: Thanks for the comment. We have made a change in the revision, accordingly.

Pg. 18 and 19. The logic behind the interpretation of these results is difficult to follow. There are more possibilities than the two options (1. FMR1 directly regulates through

RNA decay and 2. aberrant stability of mRNAs induced by loss of maternal FMR1 is a “byproduct” of translational repression) listed. Is there a competition between RNA decay and translation repression modulating the final protein levels? How can the data for Group 2 (mRNA no change, protein up) be explained? Is there a role for nuclear export of m6A labelled mRNA involving FMRP (Edens et al. reference) that might help explain these results?

Response: In the Fig. 5a of the original version, we used protein MS-Spec datasets and RNA-Seq datasets generated from *fmr1* maternal mutant embryos and wild type embryos at the 5-6 hour stage to analyze integrative changes of gene expression at both mRNA and protein levels. Actually, the purpose of Fig.5a is for detecting whether FMR1 has a role in regulating translation repression (or controlling translation rates), maternal RNA degradation or both. Our findings showed two distinct groups (group 2 and group 3), indicating that FMR1 represses gene expression largely through two different mechanisms. *Drosophila* early embryo is also known as syncytial blastoderm, which means a common cytoplasm containing many nuclei. Importantly, in *Drosophila* early embryos, FMR1 localizes in cytosolic compartment (**Fig. S8a-c in the revision**). Thus, the nuclear export of the m6A-modified mRNAs may not be a major mechanism by which FMR1 regulates embryonic development in *Drosophila*. We have cited the paper (Edens et al.) in the reference list of manuscript.

Pg. 19 “were significantly correlated with m6A modification (Fig. 5b).” What specifically is being correlating here? That m6A modifications are enriched within group 3 relative to group 2? Or to group 2 and group1? A visual inspection of the plots suggests that the phrasing here is incorrect.

Response: Yes, by performing the chi-square test, we concluded that m6A modifications are enriched within the group 3, when compared to the group 2. We have revised the manuscript, accordingly.

Pg. 21 “little is known about how maternal RNA decay is triggered and how maternal RNAs (but not zygotic RNAs)” The paper does not really clearly explain preferential

degradation of maternal RNA over zygotic RNA. Is m6A specifically enriched on maternal RNA or is there at least a delay in the methylation of the zygotic RNA?

Response: We have answered the similar question above. Our results have suggested that maternal mRNAs could be major targets for FMR1.

Does FMR1 recognize m6A modified RNA on its own or does it do this with Ythdf (see BiorXiv doi: <https://doi.org/10.1101/2020.03.04.976886> and reference this.) This paper should be cited as it discusses the role of Ythdf and ythdc1 as major m6A binders in *Drosophila*. Nat. Commun. 2021 Mar 5;12(1):1458.doi: 10.1038/s41467-021-21537-1. A neural m6A/Ythdf pathway is required for learning and memory in *Drosophila*.

Response: As mentioned above, our results from biochemical experiments, such as EMSA and SPR, suggest that FMR1 alone can bind m6A-modified RNAs in a sequence-specific manner. Importantly, we found that loss of maternal Ythdf did not affect the *Drosophila* embryogenesis.

Regarding the mentioned reference, in our study we did not detect apparent binding affinity between *Drosophila* Ythdf (dYthdf) and m6A. Indeed, we have performed additional biochemical assays to test whether dYthdf binds m6A-modified RNA (using the human Ythdf2 (hYthdf2) as a positive control). 1) the gel-shift assays showed that dYthdf has no binding affinity to multiple m6A modified RNA probes (**the Response Figure 1a, b**). By contrast, we found that hYthdf2 exhibited strong binding affinities to the probes in the same experimental setting (**the Response Figure 1a, b**). 2) The SPR analysis again showed that dYthdf has no binding affinity to the m6A modified RNA probes (**the Response Figure 1c-f**). The mentioned reference reported that probe containing “AAAmCU” probe could bind the dYthdf protein, we used both the gel-shift and SPR assay to test this possibility, and found that hYthdf2 could bind the AAAmCU, but dYthdf did not (**the Response Figure 1g-k**). Our results strongly suggest that in contrast to hYthdf2, the dYthdf has no apparent binding affinity to m6A-modified RNAs, and it is not essential for *Drosophila* early embryonic development.

This additional reference should be included:

Zhang, F., Kang, Y., Wang, M., Li, et al. Fragile X mental retardation protein modulates the stability of its m6A-marked messenger RNA targets. *Hum. Mol. Genetics.*, 27, 22, 2018.

Response: We have included the reference in the revision.

Responses to Reviewer #3's comments:

The authors test the role of methylated mRNAs and the RNA binding protein Fmr1 in embryonic development. They focus on understanding the embryonic degradation of maternally loaded mRNAs, which is a necessary step in the switch from maternal to zygotic control of embryogenesis. The authors claim that they have discovered an essential role for m6A marks in the degradation of maternal mRNAs, through the binding of methylated mRNAs by Fmr1. Furthermore, Fmr1 appears to preferentially bind to m6A-marked mRNAs, and forms particles dynamically during embryogenesis, which may indicate a dynamic function such as mRNA turnover or another function not explored by the authors.

I have major concerns about this interpretation of the data in the manuscript, which in my opinion undermines the conclusions made by the authors:

Response: We agree with that additional evidence should be provided to support the general conclusion of our manuscript. To address the concerns raised by the reviewer below, we performed a series of important experiments and made further detailed analysis.

(1) Are m6A and Fmr1 required for maternal mRNA degradation? The authors own evidence indicates that there is unlikely to be a major/direct role for either in maternal mRNA degradation.

About 1/3 of maternal mRNAs are marked for degradation during the first ~3 hours of *Drosophila* embryogenesis (De Renzis et al. 2007), a process which requires RNA decay factors such as Smaug (Benoit et al. 2009). While ~25% of degraded maternal

transcripts are cleared through strictly maternal machinery, ~75% are degraded through zygotic machinery or by both maternal/zygotic mechanisms (Thomsen et al. 2010, Tadros et al. 2007, De Renzis et al. 2007). Surprisingly, the authors observe no effect on mRNA turnover at the 2-3 hour timepoint in which most maternal mRNAs are turned over, either in mutants with reduced m6A (Figure 1D) or in Fmr1 mutants (Figure 2C). The authors only observe differences in maternal mRNA levels several hours after zygotic genome activation is normally induced. Any perturbations that reduce zygotic transcription would have the effect in RNA sequencing data of making maternal transcripts appear more abundant. Indeed a large fraction (~25%) of Fmr1 null mutant embryos fail to properly cellularize, an event intimately tied to zygotic genome activation (Monzo et al. 2006). There is no attempt by the authors to show that Fmr1 or Mettl3-14 mutants lead to specific defects in RNA turnover rather than general defects in the developmental progression of embryos. The authors may have come to the same conclusions by examining nonspecific mutants blocking normal developmental progression.

Response: Actually, in this study we did not propose that FMR1 plays major (or global) roles in regulating maternal mRNA decay. Instead, our findings suggest that FMR1 acts in conjugation with the m6A modification to regulate a portion of maternal mRNA degradation, in addition to its role in controlling the translational repression of some targets.

Regarding the reviewer's comment: "*There is no attempt by the authors to show that Fmr1 or Mettl3-14 mutants lead to specific defects in RNA turnover rather than general defects in the developmental progression of embryos.*", in fact, in this study we paid our main attention on investigating the behavior and role of FMR1 granules during the early embryonic development, since RNA binding proteins have been well-documented to regulate the fate and function of their target RNAs by forming RNP complexes/granules. Importantly, we have found that FMR1 granules undergo a process from condensation (2-3 hour) to de-condensation (5-6 hour) during early embryonic development. Based on our *in vitro* phase separation analysis showing that size of FMR1 granules can be markedly enhanced by recruiting target mRNAs (including both

m6A-modified and unmodified RNA), we argue that dynamical phenomenon of FMR1 granules in a such short timing window is a hallmark feature for FMR1 to selectively associate with targets and regulate their fate in *Drosophila* early embryos. Consistently, in the period of revision, we performed RNA-immunoprecipitation experiments using anti-FMR1 antibody followed by performing q-RT-PCR assays. We measured levels of several associated mRNAs in immunoprecipitants at the above-mentioned time points (the 2-3 hour and the 5-6 hour), and found that levels of these targets at the 2-3 hour stage is significantly higher than that in the 5-6 hour stage. Thus, our findings suggest that FMR1-mediated target maternal RNA decay is highly correlated with the dynamics of FMR1 granules (**the Response Figure 5a-f**).

FMR1 preferentially bind to m6A- modified mRNAs. Based on our *in vitro* assays, we found that loss of maternal Mettl3-14 led to a specific defect by strongly affecting the FMR1 granules condensation at the 2-3 hour stage. We also used anti-FMR1 antibody to perform RNA-immunoprecipitation followed by q-RT-PCR analysis, and found that levels of associated RNA in FMR1 immunoprecipitants from the *mettl3-mettl14* double maternal mutant embryos at the 2-3 hour stage were markedly reduced, when compared to that in the stage-matched wild-type control embryos (**the Response Figure 5a-f**). Our findings suggest that m6A modification affect FMR1 granule condensation through regulating the binding of FMR1 to its mRNA targets. Our model can explain the comment “...no effect on mRNA turnover at the 2-3 hour timepoint in which most maternal mRNAs are turned over, either in mutants with reduced m6A (Figure 1D) or in *Fmr1* mutants (Figure 2C).” raised by the reviewer.

In response to the comments of “*The authors may have come to the same conclusions by examining nonspecific mutants blocking normal developmental progression.*”, we sought to block normal developmental progression by incubating embryos under 33°C to generate nonspecific defects. We found that under this condition, ~50% of embryos undergo embryonic lethal, which is similar to that in *fmr1* maternal mutants (**the Response Figure 6a**). However, although a portion of these incubated embryos had cellularization defects (**the Response Figure 6b**), RNA-seq analysis revealed that transcriptome profile of the incubated embryos is different from

that of *fmr1* maternal mutant or *mettl3-mettl14* double maternal mutant embryos (the Response Figure 6c-e).

Response Figure 5. a-f, q-RT-PCR analysis of indicated mRNAs immunoprecipitated by anti-FMR1 antibody. Data expressed as means of 3 independent experiments. Error bars indicate mean \pm s.d.. * $P < 0.05$, ** $P < 0.01$, *** $P < 0.001$. n.s., not significant.

Response Figure 6. a, Relative hatching rate of embryos with indicated genotypes. The two-tailed Student's t test was used to analyze statistical variance. Data expressed as means of 3 independent experiments. Error bars indicate mean \pm s.d.. *** $P < 0.001$. **b**, Nuclear fallout in embryos under

33°C at stage of 9-13 cycles. Scale bar, 10 μ m. **c**, Volcano plot showing transcriptome-wide \log_2 fold changes in 25°C and 33°C incubated embryos. Upregulated and downregulated genes are highlighted in red and blue, respectively. **d**, Overlap of aberrantly up-regulated transcripts in 33°C incubated embryos and up-regulated genes of degraded maternal group in *fmr1* maternal mutant embryos. **e**, Overlap of aberrantly up-regulated transcripts in 33°C incubated embryos and up-regulated genes of degraded maternal group in *mettl3-mettl14* double maternal mutant embryos.

(2) Hundreds of mRNAs are translationally misregulated in Fmr1-deficient oocytes prior to embryogenesis (Greenblatt et al. 2018). How can the authors exclude contributions of altered levels of one or more of the hundreds of Fmr1 targets defects? It is important to note that the RNA decay factor Not1 is one such target of Fmr1 translationally upregulated by in oocytes.

Response: Our study mainly focused on investigating whether FMR1 RNP-granule condensation and de-condensation contribute to the degradation of maternal mRNAs. As mentioned above, we found that m6A modification is important for FMR1 granule condensation, thus regulating the degradation of a portion of maternal RNAs. In order to test whether Not1 influences the function of FMR1, we employed two Not1 knockdown transgenes, P{shNot1-1} and P{shNot1-2}. Using the UAS-GAL4 system, we knockdown Not1 in germ cells using the *vasa-gal4-*vp16** driver. As shown in **Response Figure 7a, b**, Not1 was significantly reduced (~90%) in the ovaries carrying the *vasa-gal4-*vp16** and P{shNot1-1}, and the Not1 knockdown embryos were not available, because the flies were sterile. Of note, Not1 was partially reduced (~40%) in the ovaries carrying the *vasa-gal4-*vp16** and P{shNot1-2}, as indicated by q-RT-PCR analysis (**the Response Figure 7c**). This reduction levels are similar to that described by Greenblatt et al. Since the flies from *vasa-gal4-*vp16**; P{shNot1-2} were fertile, we obtained embryos with weak knockdown of Not1, and further RNA-seq analysis revealed that they displayed different gene expression pattern from the *fmr1* maternal mutant did (at the 5-6 hour stage) (**the Response Figure 7d**).

Response Figure 7. **a**, q-RT-PCR experiments were used to analyze the effectiveness of *not1* knockdown 1# in ovaries. Data expressed as means of 3 independent experiments. Error bars indicate mean \pm s.d. The two-tailed Student's t test was used to analyze statistical variance. ***, $P < 0.001$. **b**, *not1* knockdown 1# flies were stained with anti-DAPI (blue) and anti-Hts (red) antibodies. **c**, q-RT-PCR experiments were used to analyze the effectiveness of *not1* knockdown 2# in ovaries. Data expressed as means of 3 independent experiments. Error bars indicate mean \pm s.d. The two-tailed Student's t test was used to analyze statistical variance. ***, $P < 0.001$. **d**, Overlap of aberrantly up-regulated genes in weak knockdown of *Not1* embryos and up-regulated genes in *fmr1* maternal mutant embryos.

(3) Many studies have been performed on Fmr1-bound mRNAs. There are no specific motifs that have emerged that are consistently highly enriched among Fmr1 target mRNAs as compared to non-targets. Multiple groups have found rather that mRNA length appears to be the major determinant in Fmr1 binding (Li et al. 2020, Sawicka et al. 2019). Is there any correlation with mRNA length and mRNA stabilization in Fmr1 or Mettl3-14 mutants at the 5-6 hour timepoint? In addition, is there any correlation between oocyte Fmr1 targets and those stabilized in Fmr1 mutant 5-6 hour embryos? If these relationships exist it would strengthen the authors' claim that FMRP is playing a direct role in the process they are studying.

Response: In this work, we initiated our study by analyzing whether the m6A modification is involved in the early embryonic development, and found that while the Mettl3 and Mettl14 methyltransferase complex is required for early embryos, the

canonical m6A reader YTH-domain containing proteins are dispensable for this process. We therefore performed the immunoprecipitation experiments using m6A-modified probe and unmodified probes and identified that FMR1 is highly enriched in m6A-modified immunoprecipitants. Several lines of evidence strongly support the argument that FMR1 regulate embryonic development by preferentially binding m6A-modified RNA in a sequence-dependent manner. First, by conducting extensive biochemical experiments, we found that FMR1 preferentially binds m6A-modified RNA in a sequence-dependent manner. Second, our genetic study suggested that FMR1 genetically interacts with the m6A pathway to regulate embryonic development. Third, RNA-seq analysis suggest that FMR1 and m6A share a set of common targets in embryos. The m6A appears to be important for FMR1 granule condensation, through which many other RNAs, including RNAs even with low binding affinity, can be efficiently recruited into FMR1 granules. This can explain why it is difficult to identify specific binding motif in FMR1-bound RNAs in the previous studies.

In response to the reviewer comment, we have examined the length of those stabilized mRNAs in *fmr1* and *mettl3-mettl14* double maternal mutants at the 5-6 hour timepoint, and found most of them have a length with 1000-3000 nt (**the Response Figure 8**).

Response Figure 8. Distributions of read density over gene length in *fmr1* and *mettl3-mettl14* double maternal mutants at the 5-6 hour timepoint.

(4) If FMRP granule formation is important for RNA decay, why is it occurring at a

timepoint in which there is no apparent difference in RNA levels in WT vs. *Fmr1* mutants? If these granules are playing a direct role, can the authors find enrichment for any mRNA supposedly targeted by *Fmr1* for degradation in these granules?

Response: Our results showed that the condensation of FMR1 granules occurs at the 2-3 hour stage, and their de-condensation occurs at the 5-6 hour stage. Because the aberrant up-regulation of a portion of maternal RNAs are mainly detected at the 5-6 hour stage, we proposed a model that condensation of FMR1 granules is important for degradation of FMR1 target maternal mRNAs, which, in turn, leads to the de-condensation of FMR1 granules, as we observed that two events are accompanied with each other. To strengthen our conclusion, we performed RNA-immunoprecipitation experiments using anti-FMR1 antibody followed by performing q-RT-PCR assays. We found that levels of these targets at the 2-3 hour stage is significantly higher than that in the 5-6 hour stage. These results suggest that the condensation status of FMR1 granules is important for the ability of FMR1 to recruit target mRNAs.

(5) In comparing Group 2 and Group 3 genes that show similar profiles in *Fmr1* mutant and mutants lacking m6A (Fig. 5), the authors make a strange finding that mRNAs that are increased in their sequencing data in 5-6 hour *Fmr1* mutants and mutants lacking m6A do not show an increase in protein abundance, whereas only mRNAs that are unchanged show an increase in protein abundance. This appears to be inconsistent with known RNA turnover mechanisms in the early embryo. mRNA decay induced by known RNA degradation factors such as Smaug and Me31B is intimately associated with translational repression. Why would the *Fmr1*/m6A-dependent degradation system operate differently from known RNA degradation pathways?

Response: Actually, we did not use mutants lacking m6A (*mettl3-mettl14* double maternal mutants) in Fig. 5 (a-b) of the original manuscript. In the Fig. 5a of the original version, we used protein MS-Spec datasets and RNA-Seq datasets generated from *fmr1* maternal mutant embryos and wild-type embryos at the 5-6 hour stage to analyze integrative changes of gene expression at both mRNA and protein levels. The purpose of Fig.5a is for detecting whether FMR1 has a role in regulating translation repression

(or controlling translation rates), maternal RNA degradation or both.

Indeed, FMR1 has been proposed as a translational repressor in many studies, but our RNA-seq analysis revealed that aberrant upregulation of a portion of maternal mRNAs in the *fmr1* maternal mutant embryos mainly occurs in the 5-6 hour stage. We can hypothesize that, if aberrant upregulation of a portion of maternal mRNAs were attributed to the defective role of FMR1 in translational repression, one would have also seen aberrant upregulation of protein products corresponding to aberrant upregulated mRNAs in *fmr1* maternal mutant, when compared to wild-type control.

However, in contrast to the above hypothesis, our findings showed that while genes in the group 2 (protein up but mRNA no significant change in *fmr1* maternal mutant) are regulated through translation repression, genes in group 3 (mRNA up but protein no significant change or down in *fmr1* maternal mutant) are regulated likely through RNA degradation. These findings suggest that FMR1 has roles in regulating both translation repression and RNA degradation.

(7) As far as I can tell there is no attempt to show that the mRNAs that are stabilized in m6A-deficient or Fmr1 mutant embryos are actually themselves methylated. This is an important point, since it would address whether the effects of methylation are direct or indirect.

Response: In response to the reviewer's comment, we analyzed our datasets, and found that some of the mRNAs stabilized in m6A-deficient and *fmr1* maternal mutant embryos are methylated in wild-type embryos. We have included these results in the revision (**Fig. S11a-c in the revision**).

(8) Finally, the authors should show using qPCR, Northern Blot, or RNA seq with proper normalization controls, that the effects they observe are due to increased levels of maternal mRNAs at the 5-6 hour timepoints rather than decreased zygotic gene expression leading to an RNA-seq normalization artifact.

Response: As the reviewer requested, we have included q-PCR results in the revised manuscript (**Fig. S11d-e, h-j in the revision**).

Responses to Reviewer #4's comments:

Figure 1e/S3d/e corresponding to RNA pull down coupled to LC-MS/MS to identify m6a interacting proteins

In Figure 1e the author show the schematic representation of RNA pull down coupled to LC-MS/MS to determine proteins that preferentially and/or specifically bind to m6a RNA. This is a nice experiment but for unclear reasons the authors fail to show any results that they have obtained.

- What kind of quantitative approach was used to determine proteins that preferentially bind to m6a RNA over unmodified RNA (and the other way around). Without a quantitative approach the authors cannot make any claims about preferential binding.

Response: We thank the reviewer for the suggestion. We performed western blot to validate the quality of proteomic data by using a few of selected proteins, including FMR1, Ote and pAbp that are available antibodies (**the Response Figure 9**). The commonly used criteria of fold change (>2) and q-value (<0.05) was further applied to define the proteins that preferentially bind to m6A. One of the key m6A binding protein, FMR1, has been identified through LC-MS/MS analysis, and this key result was confirmed by multiple other experiments, such as western blot, gel-shift assay and SPR.

Response Figure 9. Validation of some proteins by western blot.

- How many biological replicates were performed?

Response: For each LC-MS/MS analysis, three biological replicates were performed.

- The authors need to use a quantitative approach of their choice and then use a

statistical approach based on FDR and fold change to determine proteins preferentially binding to m6A RNA. I suggest showing the data in a volcano plot as usually done for quantitative proteomics data. This should be part of Figure 1.

Response: We thank the reviewer for the constructive suggestion. The m6A binding proteins have been defined as described in the first comment. To show the quantitative proteomics data, we have included a volcano plot in the revised manuscript (**Fig. S3d in the revision**).

- Were any expected proteins identified that are known to bind to m6A RNA (these ones should also then be indicated in the volcano plot)

Response: Consistent with the previous reports that Ythdc specifically bind m6A-modified RNAs³, we found that Ythdc was specifically present in m6A-immunoprecipitant. As shown in **Fig. S3d in the revision**, Ythdc has been indicated in our new volcano plot.

- A table with all quantified protein groups with quantified fold changes and FDR should be provided as supplementary dataset.

Response: All quantified protein groups with quantified fold changes and FDR (q-value) have been provided as supplementary data.

- All raw proteomics data should be uploaded to a public repository such as Pride (<https://www.ebi.ac.uk/pride/>) (from Figures 1, S3, 5 and S10)

Response: All raw proteomics data have been deposited to the ProteomeXchange Consortium.

PXD026356: <https://www.iprox.cn/page/PSV023.html?url=1624498762477HCmQ>
Password: **6to1**.

- Can the authors provide an explanation in the methods what they mean by cytosolic fraction of proteins?

Response: We have added the following description in the revised Materials and

Methods: “To identify the m6A binding proteins that involved in mRNA decay, we collected cytosolic lysates from 0–2-hour embryos of *Drosophila*, and performed Biotinylated RNA pull down assay followed by the LC-MS/MS analysis.”

- Can the authors provide explanation why known m6A binder YTHDF does not bind the m6a probe in western blot in Figure 1e?

Response: We have performed extensive biochemical assays to test whether *Drosophila* Ythdf (dYthdf) binds m6A-modified RNA. 1) the gel-shift assays showed that dYthdf has no binding affinity to multiple m6A modified RNA probes (**the Response Figure 1a, b**). Of note, in the gel-shift experiments, we used the human Ythdf2 (hYthdf2) as a control, we found that hYthdf2 exhibited strong binding affinities to the probes in the same experimental setting (**the Response Figure 1a, b**). 2) The SPR analysis again showed that dYthdf has no binding affinity to the m6A modified RNA probes (**the Response Figure 1c-f**). A recent study reported that probe containing “AAAmCU” probe could bind the dYthdf protein, we used both the gel-shift and SPR assay to test this possibility, and found that hYthdf2 could bind the AAAmCU, but dYthdf did not (**the Response Figure 1g-k**). Our results strongly suggest that in contrast to hYthdf2, the dYthdf has no apparent binding affinity to m6A-modified RNAs, and it is not essential for *Drosophila* early embryonic development.

- Which *Drosophila* dataset was used for MaqQuant analysis (i.e. number of proteins and release date)

Response: *Drosophila melanogaster* proteins annotated at Uniprot database containing 309323 sequences (uniprot_taxonomy_7215), release March 2019 was used for this analysis.

Figure S3d/e corresponding to FMR1 co-IP coupled to LC-MS/MS

Here even less information is provided compared to Figure 1. Similar questions should be addressed:

- What kind of quantitative approach was used to determine proteins that preferentially

bind to FMR1 compared to IgG. Without a quantitative approach the authors cannot make any claims about proteins interacting to FMR1.

Response: As mentioned above, the LC-MS/MS data quality have been validated for those presented in the Figure 1. For FMR1 co-IP coupled to LC-MS/MS, similar criteria of fold change (>2) and q-value (<0.05) was applied to define the FMR1-associated proteins. Most importantly, top two of the screened candidates in *Drosophila*, pAbp and Caprin (a known FMR1-binding protein) have been further validated by co-IP experiments, supporting the quality of LC-MS/MS results (**Fig. S10e, Response Figure 10**).

Response Figure 10. Coimmunoprecipitation of FMR1 with Caprin in wide-type embryos at 2–4-hour embryonic stage. The lysates were immunoprecipitated with anti-FMR1 antibody, and western blot assays were performed to detect Caprin protein.

- The authors need to use a quantitative approach of their choice and then use a statistical approach based on FDR and fold change to determine proteins co-immunoprecipitating FMR1. I suggest showing the data in a volcano plot as usually done for quantitative proteomics data.

Response: We thank the reviewer for the constructive suggestion. The commonly used criteria of fold change (>2) and q-value (<0.05) was further applied to define the FMR1-associated proteins. The quantitative proteomics data was showed in a volcano plot in the revised manuscript (**Fig. S10d in the revision**).

- A table with all quantified protein groups with quantified fold changes and FDR should be provided as supplementary dataset.

Response: All quantified protein groups have been provided as supplementary data.

- In methods section, it is not stated which antibody and what protein amount was used for co-IP and if total cellular lysates or fractionated proteins were used.

Response: For co-IP coupled LC-MS/MS, anti-*Drosophila* FMR1 antibody (Abcam, ab10299) was used. For co-IP coupled western blot, we used 10 mg protein/sample from total cellular lysates. Anti-*Drosophila* FMR1 antibody (Abcam, ab10299) and pAbp/Caprin (self-generated; validated by western blot as shown in **Fig. S10e, Response Figure 10 of the revision**) were used. We have added the information in the revised manuscript.

Figure 5A/B corresponding to protein levels in wild-type and *fmr1* mutant embryos

Here, I cannot find any information about the total proteome analysis in wild-type and *fmr1* mutant embryos either in the main text or methods section. The authors need to provide information about this. Did the authors measured total proteomes of wild-type and *fmr1* mutant embryos and again how was this done and what kind of quantitative approach was used. I could have missed that because figures seem to have a random order in the text.

Response: In the Fig. 5a of the original version, we indeed used protein LC-MS/MS datasets and RNA-Seq datasets generated from *fmr1* maternal mutant embryos and wild type embryos at the 5-6 hour stage to analyze integrative changes of gene expression at both mRNA and protein levels. We have added the description in the revised manuscript.

Figure S10 corresponding to DIA-MS analysis of embryo-derived proteins

Why was here DIA analysis used? The rationale why DIA comes into play here is not clear in the manuscript text at all. Which specific proteins were authors interested to monitor? I realize now that this is probably the data used also in Figure 5. The orders of figures need to be re-organized and it has to be clear in the main text what is shown in Figure 5.

Response: Yes, Figure S10 is the supplementary data for Figure 5. In our study, we found that loss of maternal FMR1 led to aberrant stable of a portion of the degraded maternal mRNAs. Thus, there are two possibilities that can explain our results: 1)

FMR1 directly regulates control target gene expression through RNA decay; 2) aberrant stable of maternal mRNAs induced by loss of maternal FMR1 could be a consequence (a byproduct) of defective translational repression. To distinguish these two possibilities, we performed data-independent acquisition (DIA) quantitative proteomics on the wild-type and *fmr1* maternal mutant embryos at multiple developmental stages, including 0-1 h, 2-3 h, and 5-6 h stages (each sample with three biological replicates) (**Fig. S10a**). We have revised the manuscript to make our results more clearly.

Other points

What version of the Skyline software was used for the analysis?

Response: Skyline Batch 21.1.0.146 was used for the analysis.

The figures do not flow correctly in the manuscript so it is very difficult to read the manuscript and identify the correct figure.

Response: We have carefully checked the manuscript and made changes in the revision.

I noticed that in some figures authors use non-corrected p-values in RNA-seq analysis instead of FDR/q-value.

Response: We have used q-value for all omics analysis throughout the revised manuscript.

In the methods title DIA-MS, authors also explain the DDA-MS. This has to be changed and DDA has to be included in the other paragraph

Response: We have modified the related content in the revised manuscript.

Reference

- 1 Kan, L. *et al.* A neural m(6)A/Ythdf pathway is required for learning and memory in *Drosophila*. *Nature communications* **12**, 1458, doi:10.1038/s41467-021-21537-1 (2021).
- 2 Laver, J. D., Marsolais, A. J., Smibert, C. A. & Lipshitz, H. D. Regulation

and Function of Maternal Gene Products During the Maternal-to-Zygotic Transition in *Drosophila*. *Current topics in developmental biology* **113**, 43-84, doi:10.1016/bs.ctdb.2015.06.007 (2015).

- 3 Kan, L. *et al.* The m(6)A pathway facilitates sex determination in *Drosophila*. *Nature communications* **8**, 15737, doi:10.1038/ncomms15737 (2017).

REVIEWER COMMENTS

Reviewer #1 (Remarks to the Author):

The authors answered my questions. It might be important to point out that FMRP may interact with partner proteins such as YTHDF proteins and collectively affect decay of some maternal RNA (as pointed out by reviewer 2 in a few ref.). These proteins tend to work in granules.

Reviewer #2 (Remarks to the Author):

The paper provides a very interesting contribution to furthering our understanding of the role of FMR1 and m6A in regulating RNA biology. The revised version provides additional supportive data, however there remain a number of issues. Not all statements in the paper are well supported. The data do not necessarily support a direct role for FMR1 in degradation of m6A modified RNA molecules with impact on degradation of a specific subset of maternal mRNA via a phase separation mechanism. The in vivo phase separation results in figure 4i do not show a really robust demonstration of the difference with and without mettl3-mettl14. Some points are also incorrectly interpreted (see point 2) and not enough detail is included to reproduce some experiments (see 4,5 and 8).

1. “Interestingly, we found that the m6A signal strengths for 380 degraded maternal mRNAs were decreased markedly from the 0–1-hour stage to the 5–6-hour stage (Fig. S1j), whereas no apparent difference was observed in levels of m6A modification of the 454 stable maternal transcripts between the two stages”. This actually suggests that the m6A modification is not itself the signal to degrade the maternal mRNAs, unless the authors can demonstrate some difference in the sequences that are methylated.
2. “indicating that the V311 residue is critical for preferential binding of FMR1 to RNAs containing the m6A-modified “AGACU” motif.” This statement is still incorrect as indicated in our first review. It is correct to say that the WT protein binds preferentially to the m6A modified RNA, but as we previously indicated these results do not support the statement that the V311 residue is responsible for the preferential binding. A lysine or arginine residue at the 311 position may simply enhance binding to the AGACU motif, such that both the methylated and unmethylated forms are bound with high affinity. Alternatively, the arginine or lysine residue may selectively enhance affinity for the unmethylated motif. There is no support for the valine itself being a key residue in the preferential binding of the methylated motif. The authors need to change the language to be precise.
3. Figure 2j suggests that loss of binding to methylated AGACU motifs may not be the problem that causes embryonic lethal phenotypes. Specifically, when the V311K mutant, which binds to methylated AGACU motifs with roughly the same affinity as WT (65 nM vs 80 nM), is reintroduced the embryonic lethal phenotype is not suppressed.

4. Protein concentrations for the phase separation assays, which are critical, are not listed anywhere. Therefore, it is impossible to reproduce these experiments.
5. 4h – what is the sequence of the mutant RNA? How do the authors know it does not bind to FMR1?
6. “The co-IP analysis suggested that FMR1 and pAbp could form a complex in wild-type early embryos” Doesn’t it seem more likely that FMR1 is binding to RNA molecules that have pAbp protein bound?
7. Insufficient detail on purification of recombinant GFP-FMR1FL and GFP-FMR1ΔLC is provided to reproduce these experiments. Full-length FMRP is known to be difficult to purify under native conditions. Shortened isoforms can be purified to some degree, but yields for full-length protein are typically very poor (PMID 22652662). The yields (Fig S9a) seem quite good. Was this a native preparation or were denaturing conditions used? If denaturing conditions were used, how was the protein refolded given its multi-domain structure. These are non-trivial details that need to be explained in some detail, as purification of FMR1 is typically not straightforward.

Reviewer #3 (Remarks to the Author):

I am still concerned that while there does seem to be a connection between mRNAs altered in *mettl3*-*mettl14* and *fmr1* mutants that is potentially interesting to readers, the authors substantially overstate their evidence purportedly showing a direct role of FMRP granules in mRNA destabilization. The conclusion that FMRP granules destabilize a subset of maternal mRNAs are based primarily on correlations - FMRP forming granules at a time in development preceding the degradation of a fraction of maternally supplied mRNAs, without knowledge of any of the mRNAs that are presumably contained in these granules. Also, given the absence of changes to protein levels, it is unclear what the biological relevance of altered mRNA levels are to the embryonic lethality phenotype.

The authors cannot eliminate granule formation specifically without removing all of the functions of FMRP, so it is impossible to determine the role of FMRP granules vs. other roles of FMRP, i.e. controlling the expression of hundreds of genes in oocytes prior to and during the onset embryogenesis, whose dysregulation may lead to many different phenotypes. The authors appear to rule this out for one *Fmr1* target (*Not1*), as well as non-specific effects of embryonic inviability by analyzing heat stressed embryos, but the contribution of hundreds of other FMRP targets is unknown.

Importantly, consistent with the idea that mRNA stabilization is an indirect rather than direct effect of FMRP loss, the authors find that the mRNAs stabilized in *fmr1* mutant oocytes are not characteristic of FMRP-bound mRNAs which are much longer than average (Response Fig. 8) (Li et al. 2020 PMID 32179589, Sawicka et al. PMID 31860442, Greenblatt et al. 2018 PMID 30115809). Either FMRP's binding specificity is different in early embryos as compared to the other tissues where FMRP binding has been studied, or mRNAs are stabilized in *fmr1* mutant oocytes through an indirect mechanism.

Because the authors cannot actually test the role of FMRP granules directly, it would be more accurate that the FMRP granule "phase switch" is correlated with maternal mRNA turnover, rather than FMRP granules directly regulating mRNA decay.

Other comments:

- The length analysis performed as part of the reviewer response should be included in the manuscript, since it is a simple test as to whether FMRP acts directly on its targets to destabilize them.

- For Fig. 5a, how many proteins were decreased in FMRP mutant oocytes? While it is fine to focus on subsets of the data, it seems strange to arbitrarily exclude downregulated proteins from the plot, as this information is valuable to readers.

Reviewer #4 (Remarks to the Author):

The manuscript is improved but my points regarding MS analysis were not addressed satisfactory.

Please find below my specific comments.

Response: We thank the reviewer for the suggestion. We performed western blot to validate the quality of proteomic data by using a few of selected proteins, including FMR1, Ote and pAbp that are available antibodies (the Response Figure 9). The commonly used criteria of fold change (>2) and q-value (<0.05) was further applied to define the proteins that preferentially bind to m6A. One of the key m6A binding protein, FMR1, has been identified through LC-MS/MS analysis, and this key result was confirmed by multiple other experiments, such as western blot, gel-shift assay and SPR.

This question was still not adequately addressed. Did authors use metabolic, isobaric labeling or label free quantification for performing the relative quantification in all of their MS-based proteomics experiments? Depending on which approach has been used authors need to describe the labeling approach as well as the statistical methods of how they determined the fold change and q-value.

I do not understand the validation western blot since pAbp and Ote bind equally to unmodified and modified RNA and only FMR1 binds preferentially to modified RNA.

Response: We thank the reviewer for the constructive suggestion. The m6A binding proteins have been defined as described in the first comment. To show the quantitative proteomics data, we have included a volcano plot in the revised manuscript (Fig. S3d in the revision).

In this volcano plot it seems that there are around 10 proteins that show preferential binding to modified RNA over unmodified RNA. Why were none of these limited number of proteins validated by WB but WB is only shown for Fmr1. If other proteins could not be validated, this should be stated in the main text describing the biotinylated RNA pull down.

The figure legend for figure S3d also does not make sense: "d, Volcano plot showing proteome-wide

log₂ fold changes in complexes immuno-precipitated by the unmodified and m⁶A-modified probes". There is no room for "proteome-wide" here and at least a description of what x axis and y axis present as well as the number of replicates and description of statistics to determine q-value and fold-enrichment should be added. Not only for this figure but each figure with MS-based proteomics data.

Response: Consistent with the previous reports that Ythdc specifically bind m⁶A-modified RNAs³, we found that Ythdc was specifically present in m⁶A-immunoprecipitant. As shown in Fig. S3d in the revision, Ythdc has been indicated in our new volcano plot.

At a minimum the authors could have validated Ythdc with WB in Figure 1f.

Response: We have added the following description in the revised Materials and Methods: "To identify the m⁶A binding proteins that involved in mRNA decay, we collected cytosolic lysates from 0–2-hour embryos of *Drosophila*, and performed Biotinylated RNA pull down assay followed by the LC-MS/MS analysis."

This did not address my question of what cytosolic lysate for author mean and what fractionation method was used to obtain a cytosolic lysate. Depending on the method different contaminations are expected.

Response: As mentioned above, the LC-MS/MS data quality have been validated for those presented in the Figure 1. For FMR1 co-IP coupled to LC-MS/MS, similar criteria of fold change (>2) and q-value (<0.05) was applied to define the FMR1-associated proteins. Most importantly, top two of the screened candidates in *Drosophila*, pAbp and Caprin (a known FMR1-binding protein) have been further validated by co-IP experiments, supporting the quality of LC-MS/MS results (Fig. S10e, Response Figure 10).

Response: We thank the reviewer for the constructive suggestion. The commonly used criteria of fold change (>2) and q-value (<0.05) was further applied to define the FMR1-associated proteins. The quantitative proteomics data was showed in a volcano plot in the revised manuscript (Fig. S10d in the revision).

In figure legend S10d the authors should add more information about X, y axis, how they derive q-value and fold-change, number of replicates and etc.

Same as above, this question was still not adequately addressed. Did authors use metabolic, isobaric labeling or label free quantification for performing the relative quantification in this co-IP experiment? Depending on which approach has been used authors need to describe the labeling approach as well as the statistical methods of how they determined the fold change and q-value. Validation WB cannot substitute complete lack of clarity for what has been done in MS experiments. In addition, WB in S10e does not show any negative control that would not bind FMR1 in this assay.

Responses to Reviewer #1's comments:

The authors answered my questions. It might be important to point out that FMRP may interact with partner proteins such as YTHDF proteins and collectively affect decay of some maternal RNA (as pointed out by reviewer 2 in a few ref.). These proteins tend to work in granules.

Response: We thank the reviewer's support. Regarding the issue of Ythdf, we have discussed it in the new revision.

Responses to Reviewer #2's comments:

The paper provides a very interesting contribution to furthering our understanding of the role of FMR1 and m6A in regulating RNA biology. The revised version provides additional supportive data, however there remain a number of issues. Not all statements in the paper are well supported. The data do not necessarily support a direct role for FMR1 in degradation of m6A modified RNA molecules with impact on degradation of a specific subset of maternal mRNA via a phase separation mechanism. The in vivo phase separation results in figure 4i do not show a really robust demonstration of the difference with and without *mettl3-mettl14*. Some points are also incorrectly interpreted (see point 2) and not enough detail is included to reproduce some experiments (see 4,5 and 8).

Response: According to the editor's suggestion, regarding to the mechanism of whether the phase separation of FMR1 controls maternal RNA decay, we have revised title, abstract and text to tone down our data.

For Fig. 4i, we want to point out that we have provided the quantification data (**Fig. S9g and h in the new revision**). In addition, in the revision we also provided evidence from the SDD-AGE assay showing dynamic change of high molecular weight of FMR1 in wild type and *mettl3-mettl14* maternal embryos from different developmental stages (**Fig. S9i in the new revision**).

1. "Interestingly, we found that the m6A signal strengths for 380 degraded maternal mRNAs were decreased markedly from the 0–1-hour stage to the 5–6-hour stage (Fig.

S1j), whereas no apparent difference was observed in levels of m6A modification of the 454 stable maternal transcripts between the two stages”. This actually suggests that the m6A modification is not itself the signal to degrade the maternal mRNAs, unless the authors can demonstrate some difference in the sequences that are methylated.

Response: Regarding the description of “*Interestingly, we found that the m6A signal strengths for 380 degraded maternal mRNAs were decreased markedly from the 0–1-hour stage to the 5–6-hour stage (Fig. S1j), whereas no apparent difference was observed in levels of m6A modification of the 454 stable maternal transcripts between the two stages*”, we only described the results obtained from bioinformatics assays. Logically, if only based on the data shown in Fig. S1j, we could not rule out the possibility that the m6A modification is not itself the signal to degrade the maternal mRNAs. However, loss of Mettl3-Mettl14 leads to aberrant upregulation of a portion of maternal RNAs, suggesting that the m6A modification is indeed involved in maternal RNA decay in *Drosophila* early embryos. Thus, when combined with other data shown in manuscript, our findings support a notion that actually a part of m6A-modified RNAs, but not all m6A modified RNAs, are sever as signal to degrade certain population of RNAs. In fact, in this study, we identify FMR1 as a factor that preferentially bind the m6A-modified RNAs in a sequence-dependent manner.

2. “indicating that the V311 residue is critical for preferential binding of FMR1 to RNAs containing the m6A-modified “AGACU” motif.” This statement is still incorrect as indicated in our first review. It is correct to say that the WT protein binds preferentially to the m6A modified RNA, but as we previously indicated these results do not support the statement that the V311 residue is responsible for the preferential binding. A lysine or arginine residue at the 311 position may simply enhance binding to the AGACU motif, such that both the methylated and unmethylated forms are bound with high affinity. Alternatively, the arginine or lysine residue may selectively enhance affinity for the unmethylated motif. There is no support for the valine itself being a key residue in the preferential binding of the methylated motif. The authors need to change the language to be precise.

Response: According to the reviewer's suggestion, we have changed the language to describe our data.

3. Figure 2j suggests that loss of binding to methylated AGACU motifs may not be the problem that causes embryonic lethal phenotypes. Specifically, when the V311K mutant, which binds to methylated AGACU motifs with roughly the same affinity as WT (65 nM vs 80 nM), is reintroduced the embryonic lethal phenotype is not suppressed.

Response: Our results suggested that overexpression of the V311K mutant in wild type background cause a weak embryonic lethal phenotype (**Fig. S7d and e in the new revision**), suggesting that this mutant has a dominate negative role in early embryos. Consistently, overexpression of the V311K mutant in *fmr1* maternal mutant background enhanced the embryonic lethal phenotype induced by *fmr1* maternal mutant (**Fig. 2j in the new revision**).

4. Protein concentrations for the phase separation assays, which are critical, are not listed anywhere. Therefore, it is impossible to reproduce these experiments.

Response: We have included detailed information for the phase separation assays (seen in figure legends for **Fig. 4a, f and h; Fig. S9b-e in the new revision**).

5. 4h – what is the sequence of the mutant RNA? How do the authors know it does not bind to FMR1?

Response: We have included the sequence of the mutant RNA (**Table S2 in the new revision**) and new results showing that the mutant probe failed to bind FMR1 (**Fig. S9f in the new revision**).

6. “The co-IP analysis suggested that FMR1 and pAbp could form a complex in wild-type early embryos” Doesn't it seem more likely that FMR1 is binding to RNA molecules that have pAbp protein bound?

Response: In addition to the co-IP analysis showing that FMR1 and pAbp form a

complex in early embryos, we have also performed immune-staining assays, and found that pAbp signal was partially overlapped with FMR1 positive RNP granules.

7. Insufficient detail on purification of recombinant GFP-FMR1^{FL} and GFP-FMR1^{ΔLC} is provided to reproduce these experiments. Full-length FMRP is known to be difficult to purify under native conditions. Shortened isoforms can be purified to some degree, but yields for full-length protein are typically very poor (PMID 22652662). The yields (Fig S9a) seem quite good. Was this a native preparation or were denaturing conditions used? If denaturing conditions were used, how was the protein refolded given its multi-domain structure. These are non-trivial details that need to be explained in some detail, as purification of FMR1 is typically not straightforward.

Response: In our hand, the full-length *Drosophila* FMR1 is not difficult to be purified. We have included the detailed information for purification of recombinant GFP-FMR1^{FL} and GFP-FMR1^{ΔLC} proteins (See method in the new revision).

Responses to Reviewer #3's comments:

I am still concerned that while there does seem to be a connection between mRNAs altered in *mettl3-mettl14* and *fmr1* mutants that is potentially interesting to readers, the authors substantially overstate their evidence purportedly showing a direct role of FMRP granules in mRNA destabilization. The conclusion that FMRP granules destabilize a subset of maternal mRNAs are based primarily on correlations - FMRP forming granules at a time in development preceding the degradation of a fraction of maternally supplied mRNAs, without knowledge of any of the mRNAs that are presumably contained in these granules. Also, given the absence of changes to protein levels, it is unclear what the biological relevance of altered mRNA levels are to the embryonic lethality phenotype.

The authors cannot eliminate granule formation specifically without removing all of the functions of FMRP, so it is impossible to determine the role of FMRP granules vs. other roles of FMRP, i.e. controlling the expression of hundreds of genes in oocytes prior to and during the onset embryogenesis, whose dysregulation may lead to many different

phenotypes. The authors appear to rule this out for one Fmr1 target (Not1), as well as non-specific effects of embryonic inviability by analyzing heat stressed embryos, but the contribution of hundreds of other FMRP targets is unknown.

Importantly, consistent with the idea that mRNA stabilization is an indirect rather than direct effect of FMRP loss, the authors find that the mRNAs stabilized in fmr1 mutant oocytes are not characteristic of FMRP-bound mRNAs which are much longer than average (Response Fig. 8) (Li et al. 2020 PMID 32179589, Sawicka et al. PMID 31860442, Greenblatt et al. 2018 PMID 30115809). Either FMRP's binding specificity is different in early embryos as compared to the other tissues where FMRP binding has been study, or mRNAs are stabilized in fmr1 mutant oocytes through an indirect mechanism. Because the authors cannot actually test the role of FMRP granules directly, it would be more accurate that the FMRP granule "phase switch" is correlated with maternal mRNA turnover, rather than FMRP granules directly regulating mRNA decay.

Response: We thank the reviewer's comments. According to the editor and reviewer#3's suggestions, we have revised the title, the abstract and the result sections and toned down our data.

Other comments:

- The length analysis performed as part of the reviewer response should be included in the manuscript, since it is a simple test as to whether FMRP acts directly on its targets to destabilize them.

Response: As requested by the reviewer, we have included the results of the length analysis in the revision (**Fig. S4e in the new revision**).

- For Fig. 5a, how many proteins were decreased in FMRP mutant oocytes? While it is fine to focus on subsets of the data, it seems strange to arbitrarily exclude downregulated proteins from the plot, as this information is valuable to readers.

Response: We have included the information requested by reviewer in the new revision (**Fig. S10c and Supplemental table in the new revision**).

Responses to Reviewer #4's comments:

The manuscript is improved but my points regarding MS analysis were not addressed satisfactory.

Please find below my specific comments.

Response: We thank the reviewer for the suggestion. We performed western blot to validate the quality of proteomic data by using a few of selected proteins, including FMR1, Ote and pAbp that are available antibodies (the Response Figure 9). The commonly used criteria of fold change (>2) and q-value (<0.05) was further applied to define the proteins that preferentially bind to m6A. One of the key m6A binding protein, FMR1, has been identified through LC-MS/MS analysis, and this key result was confirmed by multiple other experiments, such as western blot, gel-shift assay and SPR. This question was still not adequately addressed. Did authors use metabolic, isobaric labeling or label free quantification for performing the relative quantification in all of their MS-based proteomics experiments? Depending on which approach has been used authors need to describe the labeling approach as well as the statistical methods of how they determined the fold change and q-value.

I do not understand the validation western blot since pAbp and Ote bind equally to unmodified and modified RNA and only FMR1 binds preferentially to modified RNA.

Response: Regarding the labeling approach as well as the statistical methods, according to the suggestion, we have included the detailed information in the revised manuscript (see method in the new revision).

For the validation of mass spec data, because the pAbp and Ote were originally identified as factors to bind unmodified and modified RNA probes equally in the mass spec analysis (**Supplemental table**), the reason we performed the western blot assay was to confirm the observation.

Response: We thank the reviewer for the constructive suggestion. The m6A binding proteins have been defined as described in the first comment. To show the quantitative proteomics data, we have included a volcano plot in the revised manuscript (Fig. S3d in the revision).

In this volcano plot it seems that there are around 10 proteins that show preferential binding to modified RNA over unmodified RNA. Why were none of these limited number of proteins validated by WB but WB is only shown for Fmr1. If other proteins could not be validated, this should be stated in the main text describing the biotinylated RNA pull down.

Response: In the study, we identified 12 proteins, we have validated two proteins, FMR1 and Ythdc, which is highly abundant in the early embryos. Because Ythdc is not required for embryonic development, we focused on studying the role of FMR1 in this work. It would be interesting to test whether other proteins play a role in regulating embryonic development in future. According to the reviewer's suggestion, we have made statement in the revision.

The figure legend for figure S3d also does not make sense: “d, Volcano plot showing proteome-wide log₂ fold changes in complexes immuno-precipitated by the unmodified and m6A-modified probes”. There is no room for “proteome-wide” here and at least a description of what x axis and y axis present as well as the number of replicates and description of statistics to determine q-value and fold-enrichment should be added. Not only for this figure but each figure with MS-based proteomics data.

Response: According to the reviewer's suggestions, we have made the following changes in the revision: 1) we have changed “proteome-wide” to “interactome”; 2) we have included description for x-axis, y-axis and number of replicates; and 3) we have included the description of statistics to determine q-value and fold-enrichment.

Response: Consistent with the previous reports that Ythdc specifically bind m6A-modified RNAs³, we found that Ythdc was specifically present in m6A-immunoprecipitant. As shown in Fig. S3d in the revision, Ythdc has been indicated in our new volcano plot.

At a minimum the authors could have validated Ythdc with WB in Figure 1f.

Response: We have included the data for validation of Ythdc (**Fig. 1f in the new revision**).

Response: We have added the following description in the revised Materials and Methods: “To identify the m6A binding proteins that involved in mRNA decay, we collected cytosolic lysates from 0–2-hour embryos of *Drosophila*, and performed Biotinylated RNA pull down assay followed by the LC-MS/MS analysis.”

This did not address my question of what cytosolic lysate for author mean and what fractionation method was used to obtain a cytosolic lysate. Depending on the method different contaminations are expected.

Response: We collected cytosolic lysates by performing fraction assays (detailed information has been added in the method of the new revision).

Response: As mentioned above, the LC-MS/MS data quality have been validated for those presented in the Figure 1. For FMR1 co-IP coupled to LC-MS/MS, similar criteria of fold change (>2) and q-value (<0.05) was applied to define the FMR1-associated proteins. Most importantly, top two of the screened candidates in *Drosophila*, pAbp and Caprin (a known FMR1-binding protein) have been further validated by co-IP experiments, supporting the quality of LC-MS/MS results (Fig. S10e, Response Figure 10).

Response: We thank the reviewer for the constructive suggestion. The commonly used criteria of fold change (>2) and q-value (<0.05) was further applied to define the FMR1-associated proteins. The quantitative proteomics data was showed in a volcano plot in the revised manuscript (Fig. S10d in the revision).

In figure legend S10d the authors should add more information about X, y axis, how they derive q-value and fold-change, number of replicates and etc.

Response: According to the reviewer’s suggestion, we have added more information about x-axis, y-axis, and q-value and fold-change, number of replicates.

Same as above, this question was still not adequately addressed. Did authors use metabolic, isobaric labeling or label free quantification for performing the relative quantification in this co-IP experiment? Depending on which approach has been used authors need to describe the labeling approach as well as the statistical methods of how

they determined the fold change and q-value. Validation WB cannot substitute complete lack of clarity for what has been done in MS experiments. In addition, WB is S10e does not show any negative control that would not bind FMR1 in this assay.

Response: We have included more information about how we have performed co-IP experiment followed by mass spec. Regarding the issue of the validation of WB, we have included new results including negative control (**Fig. S10e in the new revision**).

REVIEWERS' COMMENTS

Reviewer #2 (Remarks to the Author):

The authors have addressed many of our concerns, including more details and moderating their claims to be more in line with the data. While early versions indicated a definitive role for FMR1 in preferentially binding to m6A-marked AGACU motifs and regulating maternal RNA degradation, the manuscript now states that FMR1 preferentially binds mRNA containing the m6A-marked "AGACU" motif with high affinity to contribute to maternal RNA degradation, with "contribute" being the key word now, acknowledging that recognition of m6A-mark RNAs is clearly not the whole story. As an example, in figure 2j WT FMR1 rescues the negative effect of maternal FMR1 knockout on embryonic hatching rate, but V311K FMR1, which binds to the m6A modified AGACU motifs with nearly the same affinity (65 nM vs 80 nM) does not rescue. The difference with V311K is that the unmodified RNA is also recognized, not that the m6A-marked RNA is no longer recognized. So, the function being rescued here is not the ability of FMR1 to recognize m6A modified mRNA.

With the text modifications and the inclusion of key experimental details, the paper provides valuable experimental results of interest to the scientific community due to the importance of the protein system and m6A RNA modifications. There are still some problems with the interpretation. However, this is obviously a complicated process and will require much further experimentation to address the nuances. As these further experiments are certainly beyond the scope of this manuscript, the current manuscript should be accepted.

Reviewer #3 (Remarks to the Author):

The changes that the authors have made have improved the manuscript and I have no additional comments.

Reviewer #4 (Remarks to the Author):

The manuscript now includes the information I requested but I am still of opinion that authors describe the MS data not specific enough, which leads to overstating of the results at different places. Not enough information is provided in figure legends of MS experiments.

For example, the authors need to be more specific in description of the pull down results with m6a modified probe.

Line 150-153

"We identified a number of proteins that either preferentially or specifically associated with m6A-

modified RNA probes (Fig. S3d) and found that *Drosophila* FMRP (FMR1) was highly abundant in the complexes immuno-precipitated by the m6A-modified probe (Fig. S3d).”

The authors did not identify a “number” but 12 proteins that were more enriched in the pull down with m6a modified RNA. Please also state in the main text what significance cut-off was used to identify these 12 proteins. How many bound more to the unmodified probe? This can be also added in the text if the authors wish.

Responses to Reviewer #2's comments:

The authors have addressed many of our concerns, including more details and moderating their claims to be more in line with the data. While early versions indicated a definitive role for FMR1 in preferentially binding to m6A-marked AGACU motifs and regulating maternal RNA degradation, the manuscript now states that FMR1 preferentially binds mRNA containing the m6A-marked "AGACU" motif with high affinity to contribute to maternal RNA degradation, with "contribute" being the key word now, acknowledging that recognition of m6A-mark RNAs is clearly not the whole story. As an example, in figure 2j WT FMR1 rescues the negative effect of maternal FMR1 knockout on embryonic hatching rate, but V311K FMR1, which binds to the m6A modified AGACU motifs with nearly the same affinity (65 nM vs 80 nM) does not rescue. The difference with V311K is that the unmodified RNA is also recognized, not that the m6A-marked RNA is no longer recognized. So, the function being rescued here is not the ability of FMR1 to recognize m6A modified mRNA.

Response: We agree with the reviewer's comments. Based on our biochemical and genetic evidence, we argue that the mutation of V311 changed the preferential binding affinity of FMR1 to the m6A-modified RNA and unmodified RNA, thus this mutation changed the normal function of FMR1 in regulating *Drosophila* early embryonic development.

With the text modifications and the inclusion of key experimental details, the paper provides valuable experimental results of interest to the scientific community due to the importance of the protein system and m6A RNA modifications. There are still some problems with the interpretation. However, this is obviously a complicated process and will require much further experimentation to address the nuances. As these further experiments are certainly beyond the scope of this manuscript, the current manuscript should be accepted.

Response: We thank the reviewer for the great efforts on improving our manuscript.

Responses to Reviewer #3's comments:

The changes that the authors have made have improved the manuscript and I have no additional comments.

Response: We thank the reviewer for the great efforts on improving our manuscript.

Responses to Reviewer #4's comments:

The manuscript now includes the information I requested but I am still of opinion that authors describe the MS data not specific enough, which leads to overstating of the results at different places. Not enough information is provided in figure legends of MS experiments. For example, the authors need to be more specific in description of the pull down results with m6a modified probe. Line 150-153 “We identified a number of proteins that either preferentially or specifically associated with m6A-modified RNA probes (Fig. S3d) and found that Drosophila FMRP (FMR1) was highly abundant in the complexes immuno-precipitated by the m6A-modified probe (Fig. S3d).” The authors did not identify a “number” but 12 proteins that were more enriched in the pull down with m6a modified RNA. Please also state in the main text what significance cut-off was used to identify these 12 proteins. How many bound more to the unmodified probe? This can be also added in the text if the authors wish.

Response: We thank the reviewer for the suggestion. We have changed “a number of proteins” to the exact number “12 proteins”. The cut-off with “q-value < 0.05” and “fold change > 2” were used to identify these 12 proteins. Only one protein bound more to the unmodified probe (**Fig. S3d in the new revision**).

We thank the reviewer for the great efforts on improving our manuscript. In the revised manuscript, we have modified figure legends and main text, accordingly.